

# Evolving techniques in sentiment analysis: a comprehensive review

Mahander Kumar[1], Lal Khan[2] and Hsien-Tsung Chang[3,4,5]

[1] Department of Computer Science, Mir Chakar Khan Rind University, Sibi, Balochistan, Pakistan
[2] Department of Computer Science, IBADAT Internationl University Islamabad, Pakpattan Campus, Pakistan
[3] Department of Computer Science and Information Engineering, Chang Gung University, Taoyuan, Taiwan
[4] Department of Physical Medicine and Rehabilitation, Chang Gung Memorial Hospital, Taoyuan, Taiwan
[5] Chang Gung University, Taoyuan, Taiwan

## ABSTRACT

With the rapid expansion of social media and e-commerce platforms, an unprecedented volume of user-generated content has emerged, offering organizations, governments, and researchers invaluable insights into public sentiment. Yet, the vast and unstructured nature of this data challenges traditional analysis methods. Sentiment analysis, a specialized field within natural language processing, has evolved to meet these challenges by automating the detection and categorization of opinions and emotions in text. This review comprehensively examines the evolving techniques in sentiment analysis, detailing foundational processes such as data gathering and feature extraction. It explores a spectrum of methodologies, from classical word embedding techniques and machine learning algorithms to recent contextual embedding and advanced transformer models like Generative Pre-trained Transformer (GPT), Bidirectional Encoder Representations from Transformers (BERT), and T5. With a critical comparison of these methods, this article highlights their appropriate uses and limitations. Additionally, the review provides a thorough overview of current trends, insights into future directions, and a critical exploration of unresolved challenges. By synthesizing these developments, this review equips researchers with a solid foundation for assessing the current state of sentiment analysis and guiding future advancements in this dynamic field.

## INTRODUCTION

Sentiment analysis, a branch of natural language processing (NLP), concentrates on automatically recognizing and classifying emotions and attitudes conveyed in written language. In other words, sentiment analysis emphasis is on detecting emotions and viewpoints articulated in written texts or sentiment analysis involves identifying, assessing, and categorizing emotions expressed in text data (positive, neutral and negative). It is a part of effective computing focused on evaluating subjective information like emotions (*Cambria et al., 2017*; *Chaturvedi et al., 2018*). With social media exploding, there are tons of public opinions out there. Therefore, sentiment analysis is super important for understanding what people feel about stuff like business and politics (*Sánchez-Rada & Iglesias, 2019*).

Corresponding author
Hsien-Tsung Chang,
smallpig@cgu.edu.tw

In light of this, competitive research, also known as competitor analysis, helps evaluate how well a company and its products or services perform compared to others in the market. It's important to understand how customers feel about competitors to identify strengths and weaknesses. Market research, especially using sentiment analysis, is a key way to get this information. Marketers use sentiment analysis to understand customer opinions, called the voice of the customer. By looking at customer feelings and motivations, they can improve their advertising strategies. Despite gaining popularity in the past decade, sentiment analysis has roots in the late 1990s. It allows marketers to make informed decisions about communication methods, channels, and creative strategies. Additionally, sentiment analysis provides insights into competitors' customer opinions on various factors such as price, customer service, value, features, and products. By comparing how customers perceive a brand's offerings *versus* its competitors, sentiment analysis helps brands position themselves more favorably in the market.

For this reason, over the past fifteen years or so, sentiment analysis using artificial intelligence has increasingly drawn the interest of researchers not only in English language, its also gaining interest from researcher for other languages (*Khan et al., 2021*; *Khan et al., 2022a*; *Khan et al., 2022b*; *Ashraf et al., 2022*; *Amjad, Khan & Chang, 2021a*; *Amjad, Khan & Chang, 2021b*) as well. Since 2004, it has become one of the fastest-growing and most dynamic research areas, as evidenced by the noticeable rise in scholarly articles on sentiment analysis and opinion mining (*Mäntylä, Graziotin & Kuutila, 2018*).

Sentiment analysis involves several crucial steps, including text pre-processing, feature extraction, and classification. In the pre-processing step, the initial text data is cleaned up by removing irrelevant details like stop words, symbols, and numbers. Next, the text is transformed into features using methods like Word2vec, global vectors (GloVe), fastText and term frequency-inverse document frequency (TF-IDF). During the feature extraction stage, the processed text is classified into various sentiment categories using either conventional machine learning methods such as support vector machine (SVM), naïve Bayes (NB), and logistic regression, or more sophisticated deep learning models like recurrent neural networks (RNN) and long short-term memory (LSTM).

Many studies have investigated sentiment analysis, covering topics such as opinion mining, spotting fake opinions, and evaluating online reviews (*Liu & Zhang, 2012*; *Piryani, Madhavi & Singh, 2017*). Recent surveys have outlined the current challenges and proposed new directions, classification, detecting fake reviews and opinion spam, and utilizing advanced machine learning techniques for social media analysis (*Yousif et al., 2019*; *Birjali, Kasri & Beni-Hssane, 2021*; *Soleymani et al., 2017*; *Yadav & Vishwakarma, 2020*; *Balaji, Annavarapu & Bablani, 2021*; *Hangya & Farkas, 2017*; *Rajalakshmi, Asha & Pazhaniraja, 2017*). Furthermore, a critical review of sentiment analysis frameworks has identified existing shortcomings and proposed potential interdisciplinary applications (*Bordoloi & Biswas, 2023*). Overall, the field of sentiment analysis is continually advancing and refining its methods.

To our knowledge, existing surveys on sentiment analysis usually focus mainly on supervised machine learning and lexicon-based methods, often ignoring many other techniques. While this study also addresses these approaches, it distinguishes itself by

encompassing the most commonly utilized techniques. Previous surveys have tended to prioritize machine learning, deep learning, and Lexicon-based approaches, often overlooking other sentiment analysis techniques. In contrast, this article provides a comprehensive examination, covering an extensive range of methodologies like Transformer Based Approaches. Furthermore, while other surveys may concentrate on specific tasks, challenges, or issues, like product reviews, our study provides a holistic overview of sentiment analysis, exploring various aspects including problems, tools, and methodologies. Its wide-ranging approach makes it especially useful for both researchers and beginners, bringing together a lot of information on the topic into one resource.

Our article sets itself apart from other surveys by offering a thorough examination of the pros and cons of various sentiment analysis approaches, aiding researchers in selecting the most suitable approach for their specific problems. This survey's main contributions include:

- Exhaustive review of literature to provide a detailed description of the sentiment analysis approaches and to detect commonly used tools for this task.
- Categorization of the predominant sentiment analysis approaches, accompanied by concise summaries to offer an outline of the available approaches, encompassing lexicon-based methods, machine learning, hybrids, and others.
- To stay updated on the latest research trends, it's important to summarize both the advantages and challenges of sentiment analysis.
- This involves comparing each method's pros and cons to guide the selection of the most suitable approach for specific sentiment analysis tasks. While this method facilitates informed decision-making, it also requires considering the complexities and subjective aspects inherent in such comparisons.

## Motivation

Nowadays, current literature on sentiment analysis mainly focused on supervised machine learning and lexicon-based approaches. Although these approaches are crucial, many reviews neglect other significant approaches. This study addresses this shortcoming by exploring not only the well-established methods like machine learning, deep learning, and lexicon-based approaches but also a wider range of sentiment analysis techniques like transformer based approaches.

Past literature reviews have often concentrated on conventional methods, thereby missing out on unconventional yet potentially impactful techniques. This article stands out by offering an extensive review of the full spectrum of sentiment analysis methods, including those that are new and less frequently covered. This approach provides a more comprehensive view of the field.

Our review aims to explore contemporary trends and unresolved issues within sentiment analysis. Despite a substantial amount of research, gaps remain, particularly concerning the incorporation of recent methodologies and comparative evaluations of various techniques. This review integrates recent studies to present a detailed overview of the field and to identify areas requiring further exploration.

The objective of our review is to deliver a current and in-depth analysis of text sentiment analysis, which is vital for advancing both theoretical insights and practical applications. By examining a diverse range of techniques and addressing the shortcomings noted in earlier surveys, this study seeks to pave the way for future research and development in sentiment analysis.

In conclusion, this article not only reviews established methods but also incorporates emerging techniques, providing a comprehensive perspective on the field. It aims to guide and inform future research by identifying current trends, pinpointing existing gaps, and suggesting new avenues for exploration.

## Intended audience

The key audience of this review can be named as researchers, practitioners and students who are interested in sentiment analysis. To the researchers, it presents a comprehensive discussion of the advances and challenges over the years and the new methods including lexicon based approach, machine learning approach, deep learning approach, transformer based approach, cross-domain sentiment analysis, and ambiguity and sarcasm detection. Users of other specialized fields like marketing, customer service, and teaching can also learn about the many ways that sentiment analysis tools may be used to solve identifiable problems. In the meantime, students can rely on this review to gain some understandings of the current overall outlook of sentiment analysis. In addition, the policymakers and other decision-makers may feel that it is advantageous to learn ways of how to apply the mechanism of sentiment analysis in influencing the coming policies from the analysis of the public opinions and forms of social media.

## Timeline of sentiment analysis

Originally, sentiment analysis only emerged as an interest in the middle of the 1990s when researchers looked for techniques to procedure emotions and sentiments in a text mechanically. First of all, there were rule-based approaches, aiming at the text classification as positive, negative orneutral with the help of keywords search. Later in the early 2000s there are more advanced methods such as the use of support vector machines (SVM) and naïve Bayes in sentiment analysis, thanks to the prevalence of feature tagged data such as movie reviews. It was shifted to choose features from models that can learn from data using this period of time. The next leap in performance started from the 2010s by means of deep learning techniques, especially called LSTM networks allowing for better capture of context within textual data. In 2018, transformer models like Bidirectional Encoder Representations from Transformers (BERT) and Generative Pre-trained Transformer (GPT) came into existence that minimally redefined sentiment analysis because it beats up its accuracy and flexibility of applications that could be utilized to the use of pre-trained contextually sensitive embeddings. It has only grown for several more years to the field of multilingual emotion detection and sentiment analysis with emphasis on fairness in the AI. It shows the move from the first simple paradigm based on a rule-following approach to the second more sophisticated model based on the analysis of big data samples that captures the richness of people's language.

This article is further organized as follows: 'Survey Methodology' survey's methodology used in sentiment analysis. In 'Stages of Sentiment Analysis', we explore the comprehensive process of sentiment analysis, including data collection, pre-processing, and feature extraction. 'Sentiment Analysis Approaches' provides an overview of the different techniques and methods employed in sentiment analysis. 'Application of Sentiment Analysis' discusses practical applications of sentiment analysis across various domains. 'Challenges of Sentiment Analysis' highlights the key challenges faced in this field. 'Evolving Trends and Advantages in Sentiment Analysis' identifies emerging trends and advancements. 'Performance Evaluation Parameter' examines the evaluation parameters used to assess sentiment analysis models. 'Future Directions' outlines future directions, and finally, 'Conclusion' concludes the article.

## SURVEY METHODOLOGY

To increase survey coverage, following a predefined systematic process we achieved an unbiased review of methodology. We used search engines in credible databases, some of which included IEEE Xplore, ACM Digital Library, and Google Scholar, Scopus, and Web of Science. In the current study, we used a combination of keywords and Boolean operators to identify multiple related studies on sentiment analysis. The most important and general terms like ''Dictionary-based sentiment analysis'', ''Lexicon-based sentiment analysis'', ''Machine learning sentiment analysis'' and ''Transformer-based sentiment analysis'' have been used to increase the width of the study. Sources studied were scientific articles including peer-reviewed articles, conference papers, and technical reports published between January 2000 and July 2024. It produced a total of 980 articles from the above platforms.

The selection of the articles in the present review was conducted in two steps according to the predefined inclusion criteria. Initially, title and abstracts were screened to assess journal suitability; subsequently, full text assessments were carried out on journals that passed the initial criteria. We limited our study to works with well-defined approaches, empirical findings and pioneering research on sentiment analysis. Such a narrow selection of sources left out the works that did not provide sufficient methodological analysis, so we could ensure methodological stringency in this review.

Our approach allowed for recording of the complete range of textual sentiment analysis techniques proposed in the theoretical and application-oriented literature from the ordinary lexicon-based approach to state-of-the-art transformer models. This selection enables us to focus on core trends, strength, and concern with different techniques addressing a complex view of the status of the field as well as its development. For every step of the sentiment analysis pipeline, namely data acquisition, data pre-processing, feature engineering, levels of sentiment, as well as the embedding techniques, a description of the processes is provided. Using databases such as PubMed, Web of Science, Scopus, and Computers and Electronics in Agriculture, we searched for articles according to given criteria and selected 200 articles, which provide a broad vision of the current achievements and issue in the investigated field. In addition to outlining the development of this field, this structured approach deals with

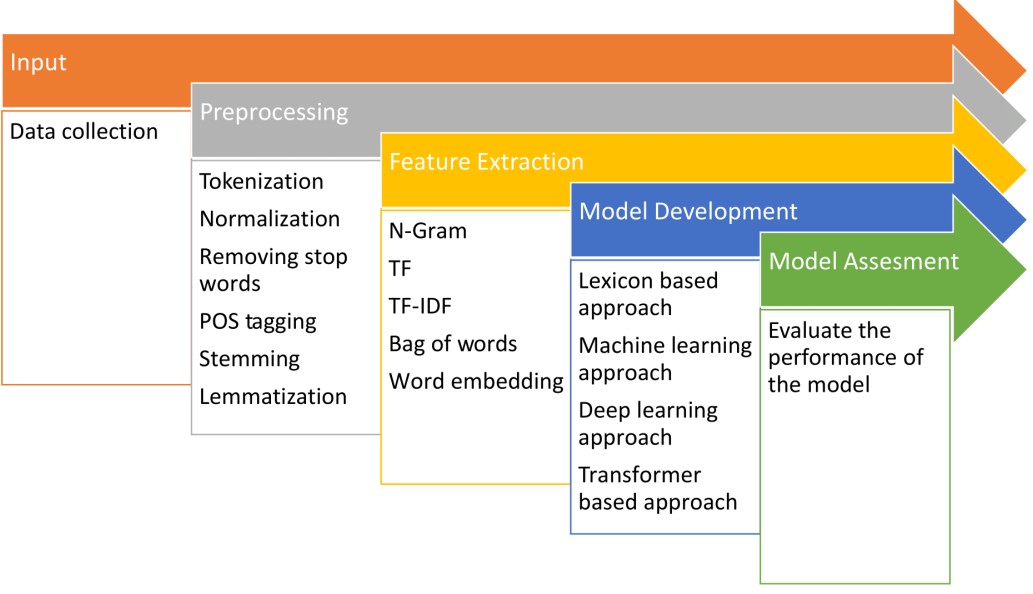

**Figure 1** **Fundamental steps for conducting sentiment analysis.**

ongoing trends and missing points, thereby providing a solid framework for subsequent research and practice.

## STAGES OF SENTIMENT ANALYSIS

The procedure for analyzing sentiment involves multiple steps. These steps include gathering data, cleaning and preparing the data, extracting relevant features, developing the model, and evaluating its performance, as depicted in the Fig. 1.

### Data collection

Data gathering can occur from various online sources, including social media platforms, web scraping, news outlets, forums, weblogs and e-commerce websites. This initial phase of sentiment analysis is the phase of text data as other data type may be included depending on the task at hand such as audio, video or location data. Key sources for data collection include:

- **Social media:** This includes data collected from social networks, especially highlight consumers' engagement with products on access, posting or sharing content. Together with that, social media data remains a vivid research object for investigating the behavior of people and their groups.
- **Forums:** Participations interact with other users in discussion and also ask questions, or make suggestions and requests through text messages in forums. These platforms yield plenty of information, useful for sentiment analysis, especially when the analysis is done within specific categories or domains.

- **Weblogs:** These consist of short entries, including viewpoints, facts, personal diaries, or links arranged chronologically. Blogs serve as valuable resources for sentiment analysis across various subjects.
- **E-commerce websites:** These platforms allow users to evaluate and express opinions about businesses or organizations. Websites such as e-commerce sites with product reviews and professional review sites provide a wealth of reviews for analysis.

In summary, data collection for sentiment analysis encompasses a diverse range of online sources, each providing valuable insights into consumer opinions and behaviors across different domains.

## Pre-processing

It has an important function of data cleaning returning more accurate results of unstructured social media data sources. It is used to remove any noise information, to decrease data content and to enhance the accuracy of classification.

- **Tokenization:** The text in the dataset is preprocessed to segment it into some coherent token values such as a tab, comma, a space or any other delimiter possible in the data.
- **Stemming:** This process normalizes each word by converting each word to it base form stripping suffixes based on certain rules of morphology .
- **Normalization:** A lot of users prefer to use abbreviations while expressing themselves or using improper punctuation. Normalization transforms those terms into their canonical written forms to allow the system to distinguish between uses of words rewritten to mean the same thing.
- **Irrelevant noise:** Usually SNS has a lot of noise regarding the data, which may hamper the efficiency of classification model. Some of the noises they include are; punctuations, numbers, symbols, username and website addresses. To reach the desired goal, all repudiated data have to be removed ahead of time in order to clean the data.
- **Stop word removal:** It is unadvisable to erase helpful words but getting rid of stopper words; words like conjunctions and prepositions which do not help to convey useful information as they are mere stop words are helpful. Using sentiment analysis (SA) or emotion recognition, stop words normally have very limited impact as to moving the sentiment from one category such as positive or happy to negative or sad respectively.

### Sentiment identification

Sentiment identification process therefore involves going through the whole document in order to determine whether voiced sentiments in the document are positive or negative. This approach entails the analyzing of sentiments using the whole document offering an overall prospect of the tenor or tone being passed across. In comparison to any approach based on words or phrases in order to perform sentiment analysis, it is more efficient when thematic elements are used as the scope of analysis because the overall sentiment of the document containing special discursive, contextual and referring signs is better described. It upgrades the process of defining a sentiment and helps to perform the analysis of textual content in further no matter the domain being belong to including, but not limited to, reviews and feedback and social media posts. These are some of the tiers of sentiments

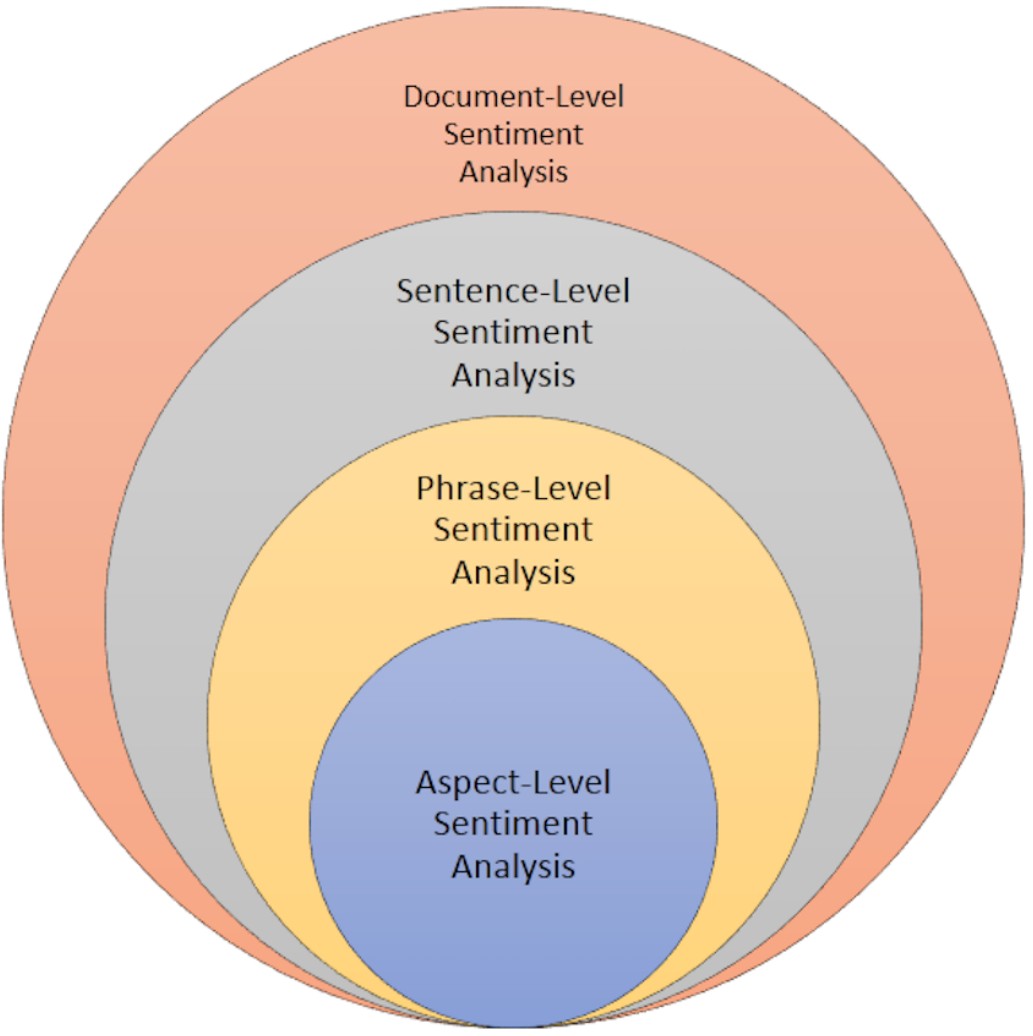

**Figure 2** **Different levels and its approaches for sentiment analysis.**

identification. It has been studied at aspect level, phrase level, and at sentence level and, at document level. Every single level, such as document, sentence, phrase, and aspect, has its own approach to sentiment analysis as shown in Fig. 2.

- **Document level sentiment analysis.** This stage involves categorizing despite a complete opinion piece conveys a favorable or unfavorable sentiment. For instance, in the case of a product review, the system discerns whether the review predominantly reflects a negative or positive viewpoint regarding the product. This process is often referred to as document level sentiment classification. At this level of examination, each document is assumed to present opinions on only one entity, such as a single product. Consequently, this method is not suitable for documents that evaluate or compare multiple entities (*Pang, Lee & Vaithyanathan, 2002*; *Turney, 2002*; *Pu, Wu & Yuan, 2019*). Sentiment identification process thus entails reviewing an entire document in order to establish if the voiced sentiments within the document are positive or negative. This approach

involves the analyzing of sentiments based on the whole document, which then provides an overall prospect (*Rao et al., 2018*) of the tenor or tone being passed across. As compared to any type of word- or phrase- based approach to sentiment analysis, using thematic elements as the scope of analysis gives a more accurate impression of the overall sentiment of the document, including all its specific contextual and discursive cues and references (*Bhatia, Ji & Eisenstein, 2015*). This integrated approach improves the process of identifying the sentiment and makes it easier to conduct subsequent analysis on textual content irrespective of domain including, but not limited to, reviews and feedback and social media posts. These are some of the tiers of sentiments identification. Sentiment analysis has been performed at various level including aspect level, phrase level, sentence level and at document level (*Saunders, 2021*).

- **Sentence level sentiment analysis.** Identifying the sentiment polarity of a sentence or particular part of text involves a decision on whether the tone of the sentence contains positive, negative or neutral sentiment and is particularly useful when conveying a sentiment in a particular text is complex and may vary greatly from area to area in the text corpus. This becomes very useful when a document contains multiple sentiments that may be across the polarity (*Yang & Cardie, 2014*; *Meena & Prabhakar, 2007*; *Shirsat, Jagdale & Deshmukh, 2019*). At this stage the goal is to evaluate sentences and determine if they have positive, negative or neutral stance. Neutral often means no opinion on that matter as well. This analysis level is directly related to the subjectivity categorization suggested by *Wiebe, Bruce & O'Hara (1999)*, where the categorization is done between explicative sentences, which are diction that may create real content or meaning and that can be referred to as objective sentences and those that represent opinionated points of view and beliefs that are referred to as subjective sentences. One has to admit that only some sentences stating opinions can be considered subjective, while there can be a lot of objective sentences containing opinions. This classification level is associated with the subjective differentiations (*Rao et al., 2018*). As in all previous cases, the sentiment polarity of each sentence will be analyzed individually, however, due to increased computation power and newly gathered training data the analysis will be performed in the same manner as in the document level approach. These individual sentence polarities can then be added up in order to determine the document polarity, or they can simply be looked at separately. Sometimes, the use of SDA at the document level may not be adequate enough for some contextual analyses as pointed out by *Behdenna, Barigou & Belalem (2018)*. In prior studies, it has mainly concerned with identification of subjectivity at the sentence level. However, addressing more challenging tasks, such as handling conditional sentences or uncertain statements, highlights the importance of sentence-level sentiment analysis in such scenarios (*Ferrari & Esuli, 2019*). Researchers have also examined clauses, yet the clause level analysis remains insufficient (*Wilson, Wiebe & Hwa, 2004*).

- **Phrase level sentiment analysis.** Phrase level analysis is the method of determining the sentiment polarity (positive, negative, neutral) of individual phrases or expressions within text. Textual phrases can be either multi-faceted, expressing multiple aspects or uni-faceted as most relevant in multi-line product reviews finding (*Thet, Na & Khoo,*

*2010*). The issue has attracted much attention among scholars in the past few years. While document-level analysis targeted at categorizing whole documents as either subjectively negative or positive, sentence level analysis provides great advantages, given that documents often contain both positive and negative sentiments. At the essence of language, the word serves as the fundamental unit, with its polarity interconnected to the subjectivity of the surrounding sentence or document. Sentences incorporating adjectives are more likely to be subjective in nature (*Fredriksen-Goldsen & Kim, 2017*). Moreover, the selected term for expression reflects various demographic attributes like age, gender, personality, social status, as well as traits, and other social and psychological characteristics, thereby serving as the cornerstone for sentiment analysis of text (*Flek, 2020*).

- **Aspect level sentiment analysis.** Aspect level sentiment analysis involves evaluating the sentiment polarity of specific aspects or features within a given context or entity. In aspect level sentiment analysis, every sentence can encompass multiple aspects, prompting a thorough examination of each aspect within the sentence to assign polarity individually. Subsequently, an overall sentiment for the entire sentence is derived by aggregating the sentiments of its constituent aspects (*Schouten & Frasincar, 2015*; *Lu et al., 2011*). Fine-grained analysis is conducted at the aspect level, previously referred to as the feature level in the term of feature-based opinion mining and summarization (*Hu & Liu, 2004*). Unlike examining linguistic builds such as phrases, clauses, sentence, paragraphs and documents, the aspect-level directly focuses on the opinion itself. This approach operates on the premise that an opinion comprises both sentiment (negative or positive) and a target. An opinion lacking identification of its target offers limited utility. Recognizing the significance of opinion targets also enhances our comprehension of the sentiment analysis challenge. For instance, in the statement "In this restaurant, the food is very delicious but the service is very poor", two aspects are evaluated: food quality is deemed "Good" (positive) while restaurant service is labeled "Poor" (negative). This approach enables the production of a well-organized summary of opinions regarding entities and their specific aspects, converting unstructured text into organized data usable for qualitative and quantitative analyses. As well as sentence-level classifications and document-level pose significant challenges, with aspect-level analysis being particularly demanding due to several sub-problems. Adding to the complexity, there exist two types of opinions (*Jindal & Liu, 2006*), regular opinions, which express sentiment on a specific entity or aspect (*e.g.*, "Pizza is very tasty"), and comparative opinions, which evaluate multiple entities by considering shared aspects (*e.g.*, "Pizza tastes better than Noodles"), expressing a preference.

### Feature selection

This layer is optionally charged with the role of identifying opinions and sentiments of each given sentence. This layer comes into force if a given sentence is a subjectivity, that is, if it gives the author's opinion about something, it classifies it, for example, as positively charged or negatively charged. This step entails a recognition of any subjectivity, and then categorization of the sentence depending on the sentiment imposed. Through making

this break down of the objective and subjective sentence, it is easier to classify opinions hence a better understanding of sentiment in the text (*Wilson, Wiebe & Hoffmann, 2005*). Before constructing a classification model, it is significant to mention that it invokes the specification of relevant features within the set data (*Ritter, Mausam Etzioni & Clark, 2012*). As a result, during the model training phase, a review can be broken down to its constituent words and included into the feature vector. When used for individual words, the method is called "uni-gram" if for pairs of words it is termed "bi-gram" and if for three-words constructions the term used is "tri-gram". Categorizing the data using both uni-grams and bi-grams provides higher levels of effectivity in comparison to the results obtained from the more established methods of analysis, mainly because they allow us to capture pure contextual data, which leads to improved overall accuracy of the results (*Razon & Barnden, 2015*). Emojis, pragmatic cues, punctuation marks and words containing slang and so on are the most favorite aspects to be chosen in the course of the sentiment analysis.

- Emoticons act as illustrations of faces accompanying emotion through visual imagery. These emotive symbols, which are grouped according to positive and negative sentiments, are central to determining success of a message at portraying the intended mood. Thus, reflecting the positive and negative feelings within the whole range from happiness and sadness to rage, –one can translate the meanings of the words lying in the middle of the scale. Positive emoticons include; love, joy while negative include; sorrow, depression, anger among others (*Tian et al., 2017*).
- Pragmatic attributes highlight the practical use of language over theoretical principles. Pragmatics examines the interplay between context and interpretation in linguistics and allied fields. It delves into concepts like implicature, speech acts, relevance, and discourse.
- Punctuation marks like exclamation points emphasize the emotion in statements. Apostrophes and question marks also play crucial roles in punctuation. Slang terms such as 'lol' and 'rofl' inject humor, especially in opinion tweets, indicating sentiment analysis.
- Slang in words like 'lol' and 'rotf' typically insert humor into a statement. Considering the nature of opinion tweets, it's reasonable to assume that slang expressions within the text can indicate sentiment. In such cases, the slang term's meaning should be substituted for analysis.

## Feature extraction

One of the most important tasks in sentiment classification is feature extraction that helps to extract helpful information from text data which directly impacts the performance of a model. For the most part, this process is emphasised in order to work shorter outlines that encase significantly more text than seemingly necessary. The feature extraction process accepts text input and produces extracted features in various formats such as Lexico-Syntactic or Stylistic, Syntactic, and Discourse-based (*Das & Bandyopadhyay, 2010*; *Abbasi, Chen & Salem, 2008*). Due to these constraints, *Venugopalan & Gupta (2015)* utilized additional features, knowing that it is difficult to extract information from text. During a pre-processing phase punctuation is removed than the text is converted to lower-case.

Nonetheless, as seen in feature extraction step described below they used punctuation but no hash tags and emoticons., as listed below.

### Term frequency

It is a method that entails tallying the occurrence of every term (word) within a document or dataset. It serves as a fundamental approach to depict the connotation of a term within a document. This method examines individual words, known as uni-grams, or combinations of two or three words, referred to as bi-grams and tri-grams, by counting their occurrences, thus serving as features (*Sharma et al., 2013*). The presence of a term assigns it a binary value of either 0 or 1. On other hand term frequency is represented as an integer value, indicating the count of its occurrences within the document under consideration.

### Term frequency-inverse document frequency

TF-IDF assesses the significance of a word within a document relative to its relevance across a corpus of documents. It helps to identify terms that are common in a specific document but uncommon across the entire collection of documents, thus providing more meaningful features.

### Parts of speech tagging

Parts of speech (POS) tagging allocates a linguistic category (*e.g.*, noun, verb, adjective) to every word in a given text. This process proves beneficial for sentiment analysis as it discerns the grammatical function of words, particularly adjectives and adverbs, known to convey sentiment. Tokens undergo classification into various grammatical categories such as verbs, adverbs, nouns, pronouns, prepositions and adjectives. For example, the phrase ''This mobile is amazing'' could be annotated as follows (*Straka, Hajic & Straková, 2016*): ''This'': determiner, ''mobile'': noun, ''is'': verb, ''amazing'': adjective. In sentiment analysis, adjectives are particularly valuable as they often express the sentiment of opinions. Part-of-speech (PoS) taggers, such as those provided by NLTK or Spacy, can be employed for this purpose. Among these, the Stanford PoS-tagger is frequently utilized in research endeavors (*Weerasooriya, Perera & Liyanage, 2016*).

### Sentiment lexicons

Specifically, sentiment lexicons or dictionaries helps in assigning sentiment scores to the words of a text by having words labeled with sentiment polarity (positive, negative or neutral sentiment). A sentiment lexicon is a list of words that are marked with their sentiment polarity, that is, as being positive or negative (*Medhat, Hassan & Korashy, 2014*), just like positive sentiment words consist of ''wonderful'', ''beautiful'', and ''amazing'', in contrast negative sentiment words encompass ''awful'', ''poor'', and ''bad''.

### Dependency parsing

A possible English definition for dependency parsing includes pondering of how words in a given word string depend on other words. This technique helps to obtain features such as syntactic patterns or cooccurrences that yield sentiment-bearing words (*Di Caro & Grella, 2013*).

### Emotion lexicons

As with sentiment lexicons, emotion lexicons are collections of words each of which are associated with a set of emotions (*e.g.*, joy, anger, sadness) (*Khoo & Johnkhan, 2018*). There are two issues that are inherent with extracting features: identifying emotionally charged words in the text and corresponding affective appraisal.

### Semantic role labeling

Semantic role labeling (SRL) determines the relation of linguistic elements to the predicate or, in other words, the roles of the words in relation to the predicate (actor, object, subject, *etc.*) (*Mohammad, Zhu & Martin, 2014*). It provides information concerning the attitude expressed by people in the direction of different object or concept in a sentence.

### Bag of words

Bag of Words (BoW) method remains as a simple procedure of extracting textual features from a reference document. For a particular document, it describes the word frequencies distinguishing between sentences as vectors based on the word dictionary (*Zhang, Jin & Zhou, 2010*). However, it has one significant disadvantage in that the resulting representations are not sensitive to the syntactic structure of the text at all.

### Word embedding

Word embeddings is the translation of words into vectors, with meaning comparisons done with structures trained in a way similar to that of neural networks. SG and CBOW algorithms adopt window type methods to predict either context from the word or words from the context. They are supposed to derive meanings within the local context useful in operations such as sentiment analysis.

There are primarily four types of text embeddings: They also classified it into four categories, namely word-based embeddings, phrase-based embeddings, sentence-based embeddings and document-based embeddings. There are some significant subcategories of these four groups mentioned above.

- **Word2vec** (*Mikolov et al., 2013*): Word2vec, a 2-layer neural network, is tasked with vectorizing tokens, is known as one of the most popular methods developed . This method comprises two primary models: SG and CBOW. Unlike CBOW, where an intention word is expected from the context words, it is predicted for the intention word with SG using context words. Notably, the observation from the evaluation results is that the proposed SG model performs better especially when large datasets are used.
- **GloVe** (*Pennington, Socher & Manning, 2014*): Global vectors for word representation (GloVe), designed by *Pennington, Socher & Manning (2014)*, is learnt from the co-occurrence matrices that show frequencies of different words within a corpus. Another advantage to its use is its ease, especially in the training of its model which benefits immensely from parallel implementation, according to *Al Amrani, Lazaar & El Kadiri (2018)*.
- **FastText** (*Mikolov et al., 2013*): FastText, developed at Facebook Artificial Intelligence Research (FAIR), Open Archives Initiative (OAI) is used for word vectorization, and word classification, and generating word embeddings. Using a linear classifier, it has a

fast model training rate, and as evidenced by *Bojanowski et al. (2017)*. CBOW as well as SG model is manageable well in FastText in order to analyze the particulars of semantic similarity.

- **ELMo** (*Ulčar & Robnik-Šikonja, 2022*): ELMo outperforms other contextually less intuitive techniques such as LSA and TF-IDF, the problems with which have been discussed above (*Peng, Yan & Lu, 2019*). It acquires word embeddings with regards to the contextual usage in the vicinity and adds more information about context. In detail, during pretraining, ELMo is able to handle polysemous word meanings in different contexts and gives the reader more profound disambiguation of the text at the later semantic level (*Ling et al., 2020*).

- **CBOW** (*Zhang, Jin & Zhou, 2010*): Another model that was proposed by *Zhang, Jin & Zhou (2010)* is Continuous Bag of Words (CBOW) which predicts a target word within the CBOW architecture relying on the context surrounding it, without paying attention to the order of the context words. On the other hand, the skip-gram model is used to predict the context words given the current word. In contrast to CBOW, skip-gram models take longer training time, and in general are more effective in handling the out of vocabulary words.

- **Skip-gram** (*Ma, Peng & Cambria, 2018*): According to *Ma, Peng & Cambria (2018)*, Skip-gram (SG) is an often used method which was derived from n-grams, which are employed in language modelling and other linguistic activities. In contrast with regular n-grams where the constituents which form it (normally words) are placed adjacently, skip-grams provide for spaces between the words. This approach aids in solving the data sparsity issue that is typical with the normal n-gram models.

- **Sentiment-Specific Word Embedding (SSWE)** (*Tang et al., 2014*): *Tang et al. (2014)* built a novel model called SSWE which is an extension of the continuous word representation incorporating sentiment cognition. The model works with positive and negative n-grams, positive sentiment values ranging between 1 and 0, negative values between 0 and 1. SSWE covers coherent sentiment expressions and does not limit to the emotional tone of verbs but also their roles and meanings.

- **GLoMo** (*Yang et al., 2018*): GLoMo, short for Graph from Low-level Modeling, is on latent graph training with an unsupervised learning strategy. This model improves abilities of diverse NLP tasks like sentiment analysis, linguistic inference, question answering, and image classification by providing an access to context possession during the training process.

- **Sent2Vec** (*Agibetov et al., 2018*): In order to address this, I will focus on the final model which is Sent2Vec. This is an unsupervised version of fastText that turns entire sentences into vectors. It employs entire phrases as context while assuming all vocabulary words as potential class labels which make it very efficient for embedding of particular sentences.

- **OpenAI Transformer** (*Liu et al., 2019*): The OpenAI Transformer model utilizes a language model as a training signal to train a large transformer framework by unsupervised learning. It offers broad applicability to a variety of tasks due to its efficiency of fine-tuning with comparatively small quantities of supervised data while being adequately specialized in its given task.

- **BERT** (*Gao et al., 2019*): Bidirectional Encoder Representations from Transformers (BERT), learn about the unlabelled text data simultaneously in both left and the right context. This makes it possible tuning to solve several NLP problems by placing on top of the pre-trained model a final layer of output.
- **Context2Vec** (*Liang et al., 2019*): The latest among the text embedding methods is Context2Vec which aims to model contextual information in a sentence. Still, it is efficient in explaining the meaning of context but not very efficient in terms of computational costs since it deals with different complex contextual details.
- **Universal Language Model Fine-tuning** (**ULMFiT**) (*Nithya et al., 2022*): ULMFiT is a text-to-text pre-trained language model that could be used for general-domain learning and then subsequently adapted to particular task domains. The method is scale, quantity and label type invariant providing natural flexibility for multiple document analysis tasks. ULMFiT employs a single architecture as well as the learning procedure to solve various problems without having to require specific domain tag.

## SENTIMENT ANALYSIS APPROACHES

There are five primary methods commonly employed for sentiment analysis; the Lexicon based approach, the machine learning approach, deep learning approach, Transformer based approach and hybrid approach as shown in Fig. 3. Moreover, scholars are consistently endeavoring to discover improved methodologies to achieve this objective with heightened precision and reduced computational demands.

### Lexicon based approach

The Lexicon-based approach relies on a sentiment lexicon containing data on words and phrases categorized as either positive or negative (*Nanli et al., 2012*). This lexicon is a compilation of linguistic features labeled according to their semantic orientation.

Initially, researchers construct a sentiment lexicon by aggregating sentiment word lists using methods like manual compilation, lexical analysis, and corpus examination.

Lexicons are tokens assigned a score indicating the neutral, positive, or negative nature of a text. The score can be determined by either polarity or intensity (*Kiritchenko, Zhu & Mohammad, 2014*). In the Lexicon based approach, each token's scores are calculated separately, and the overall polarity is assigned based on the highest individual score. The document is divided into single-word tokens.

Lexicon-based sentiment analysis is a feasible, unsupervised technique for feature and sentence level sentiment analysis, however, its main drawback is domain dependence, since words can possess multiple meanings and interpretations, causing negative outcomes in different domains. One advantage of the Lexicon-based technique is that it doesn't crave any training data, and some experts even refer to it as an unsupervised approach (*Yan-Yan, Bing & Ting, 2010*). According to *Moreo et al. (2012)*, the primary drawback of the Lexicon-based method is its strong domain orientation, which prevents terms from one domain from being utilised in another. Take the term big, for example. Depending on the context, it might have a positive or negative meaning.

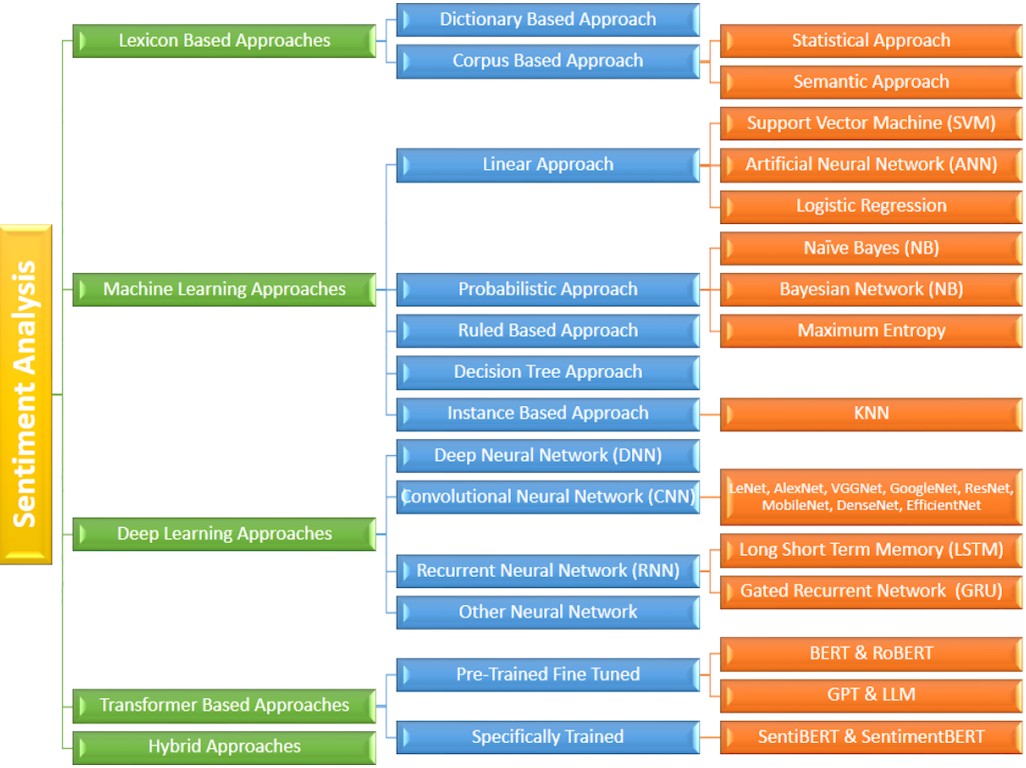

**Figure 3** Overview of various approaches to sentiment analysis, including lexicon-based, machine learning, deep learning, transformer-based, and hybrid methods.

SentiWordNet (*Sebastiani & Esuli, 2006*) and Valence Aware Dictionary and Sentiment Reasoner (VADER) (*Hutto & Gilbert, 2014*) stand out as widely-used lexicons in sentiment analysis. *Jha et al. (2018)* sought to expand the lexicon's applicability across various domains by developing a dictionary of sentiment called Hindi Multi-Domain Sentiment Aware Dictionary (HMDSAD) for analyzing sentiment at the document level. This dictionary facilitates the categorization of reviews into negative and positive sentiments. Their approach demonstrated a 24% increase in word labeling compared to the conventional general lexicon Hindi Sentiwordnet (HSWN), which is domain specific. Notably, it enhances sentiment classification performance by not only considering individual words but also exploring the semantic relationships between them, a feature lacking in conventional lexicons. Building upon this notion, *Viegas et al. (2020)* revised the lexicon by incorporating additional terms identified through word embeddings. These embeddings enabled the automatic derivation of sentiment values for new terms based on the proximity to existing words in the lexicon.

### Dictionary based approach

A dictionary systematically organizes words from a language, while a corpus comprises a varied assortment of texts in a language. This distinction is relevant to the corpus based approaches and dictionary based approaches in sentiment analysis. The dictionary

based method involves creating and maintaining a dictionary of seed words (*Schouten & Frasincar, 2015*). At first, a small collection of sentiment words, which may also include brief contexts such as negations, is assembled along with their polarity labels (*Bernabé-Moreno et al., 2020*). This dictionary is then expanded by adding synonyms (same polarity) and antonyms (opposite polarity). The effectiveness of sentiment analysis through this method hinges on the algorithm chosen. Nevertheless, However, one serious limitation of utilizing a dictionary based approach for aspect extraction is its inability to capture domain and context-dependent variations in the lexicon of opinion words.

### Corpus based approach

This approach using a predefined set of sentiment terms and their orientations as a starting point, this method finds sentiment tokens and their orientations across a sizable corpus by analyzing syntactic or related patterns. This method utilizes semantic and syntactic structures to determine the emotional tone of a sentence. It starts by using a pre-determined list of sentiment terms along with their polarity, and then analyzes sentence structures or alike model to identify sentiment words and their polarity within a large dataset. This technique is tailored to specific situations and relies heavily on annotated data for training. Nevertheless, it effectively addresses the challenge of opinion words having varying orientations depending on context. *Cho et al. (2014)* addressed contextual polarity directly to enhance the adaptability of dictionaries across diverse domains through a data-centric approach. Their approach involved three main steps: consolidating different dictionaries, eliminating words that have minimal impact on classification, and adjusting sentiment based on the particular domain being analyzed.

There are two types of corpus based approaches; statistical approach and semantic approach as defined below:

- **Statistical approach:** Statistical approaches are used in sentiment analysis, including detecting manipulated reviews through training tests. *Hu et al. (2012)* found that nearly 10.3% of products experienced management of online reviews. Latent semantic analysis (LSA) is another statistical method used to uncover relationships between documents and tokens to extract significant patterns. *Cao, Duan & Gan (2011)* utilized LSA to analyze semantic qualities in reviews from CNETdownload.com, identifying factors affecting helpful voting patterns and identifying reasons for low helpful votes.
- **Semantic approach:** In this methodology, the similarity measure is computed among tokens utilized for sentiment analysis, typically employing WordNet. This facilitates the identification of antonyms and synonyms, as words with similar meanings tend to have positive scores or higher values. *Maks & Vossen (2012)* proposed employing a semantic approach across numerous applications to construct a lexicon model for describing nouns, verbs and adjectives suitable for sentiment analysis. They delved into character-level subjectivity relations within statements, discerning unique attitudes for each character. Subjectivity is labeled based on information that includes both the identity and the stance of the attitude holder. Additionally, *Bordes et al. (2014)*, *Bhaskar, Sruthi & Nedungadi (2015)* and *Rao & Ravichandran (2009)* explored the WordNet dataset in

their respective studies, observing instances where viewer and actor subjectivities could be distinguished (*Hershcovich & Donatelli, 2021*).

## Machine learning approach

Machine learning algorithms have the capability to classify sentiments. Sentiment analysis involves recognizing and measuring the emotional tone of audio or text by employing natural language processing, text analysis, computational linguistics and similar methodologies. Within machine learning, there are two main approaches to conducting sentiment analysis:

- Unsupervised machine learning
- Supervised machine learning

Unsupervised machine learning: As the term suggests, unsupervised machine learning is an algorithm that operates on data that has not been given explicit instructions for defining categories or labels (a sort of model without supervision). These methods are based on self-learning and have demonstrated their effectiveness in natural language processing, especially in the area of sentiment classification (*Al-Ghuribi, Noah & Tiun, 2020*; *Yadav & Vishwakarma, 2020*). The majority of contemporary unsupervised sentiment classification approaches adopt a two-step procedure (*Rothfels & Tibshirani, 2010*; *Dai et al., 2021*). Initially, the sentiment intensity of the text is assessed by evaluating the strength of words and phrases that express emotions. Subsequently, sentiment categorization is executed by contrasting the determined sentiment intensity with a baseline value of '0', It does not write down rules to follow, instead it learns the patterns, structures and relationship that exists in the data.

- Clustering: The algorithm will group the similar data points together; *e.g.*, K-means clustering
- Dimensionality reduction: It simplifies large datasets and preserves the important information, such as principal component analysis This flexibility and insight without reliance on labeled examples make this method especially suited to tasks like anomaly detection, exploratory data analysis, and discovering hidden patterns.

Supervised machine learning: This method usually involves dividing the dataset into two sections; the training set and the testing set. The training set is used to teach the model by exposing it to different item features. After the model is trained, its performance is measured using the testing set, which helps determine how well the model can generalize to new data. In sentiment analysis, machine learning algorithms predominantly belong to the supervised classification category. Various algorithms utilized for sentiment classification might encompass naïve Bayes, Bayesian network, artificial neural network (ANN), support vector machine (SVM), Maximum entropy, decision trees, each offering its own set of benefits and drawbacks. Various authors have explored multiple machine learning techniques for sentiment analysis which are described as bellow.

### Linear approach

A linear approach refers to a statistical method for sentiment classification that employs linear or hyperplane decision boundaries. Typically, the term "hyperplane" is utilized when there are multiple classes involved. This method entails the use of a linear predictor, denoted as

$$p = A.X + b. \tag{1}$$

In this context, vectors A and X represent the linear weights (coefficients) and the word frequency in the document, respectively. Predictions are generated by computing the dot product of A and X and then adding a bias term, b. Linear classifiers, sometimes referred to as deterministic classifiers, are uncomplicated and can achieve excellent performance when the appropriate features are used. Here are some sub-approaches of the linear approach.

- **Support vector machine:** SVMs stand out among non-probabilistic supervised learning methods, often applied in classification tasks. SVM aims to identify the optimal hyperplane to neatly divide data into separate classes. Primarily geared towards binary classification, SVM shines particularly in handling large datasets. Notably, when it comes to text classification, among various kernel functions employed in SVM, the linear kernel proves most effective (*Mullen & Collier, 2004*). This kernel function is denoted as follows.

$$k(x_i, x_j) = x_i^T x_j. \tag{2}$$

In a study conducted by *Gamon (2004)*, a support vector machine was utilized to analyze 40,884 customer feedbacks obtained from surveys. Through experimenting with different combinations of feature sets, the researchers attained an accuracy rate reaching 85.47%. Similarly, *Ye, Zhang & Law (2009)* investigated sentiment and reviews regarding seven prominent destinations in the USA and Europe sourced from yahoo website. Employing SVM, Naïve Bayes and N-gram model, they obtained an accuracy level of up to 87.17% utilizing the n-gram model. *Sabir, Ali & Aljabery (2024)* analyzed that sentiment analysis on ChatGPT-labeled tweet datasets from Kaggle, employing five different machine learning algorithms. Three feature extraction techniques were assessed: positive/negative frequency, TF-IDF and bag of words. The highest accuracy, 96.41%, was achieved with the SVM classifier utilizing TF-IDF. *Suresh Kumar et al. (2024)* used rigorous feature selection and a multiclass HSVM classifier for text categorization, enhanced by decision trees and sentiment dictionaries expanded with Stanford's GloVe tool. Weight-enhancing methods were introduced to improve sentiment analysis accuracy, categorizing sentiments as positive, negative, or neutral. The approach achieved a 92.78% accuracy, with positive and negative sentiment rates of 91.33% and 97.32%, respectively. *Makhmudah et al. (2019)* conducted research focused on analyzing sentiment in Tweets regarding homosexuality in Indonesia. Their study introduced the application of support SVM technique for this purpose. The dataset underwent preprocessing steps including stop word removal, stemming and lemmatization; and was depicted using TF-IDF features. Remarkably, the SVM method yielded outstanding performance, achieving an accuracy of 99.5% on the dataset.

- **Artificial neural network:** It functions by extracting features through linear combinations of input data and expresses the output using a nonlinear function based on these features (*Yang & Chen, 2017*; *Moraes, Valiati & Neto, 2013*). A standard neural network typically consists of three layers: input, output, and hidden layers, each containing numerous organized neurons. Connections between consecutive layers are established through links connecting neurons from each layer. Each link is associated with a weight value, determined by minimizing an overall error function through a gradient descent training process (*Moraes, Valiati & Neto, 2013*). In their work, *Chen, Liu & Chiu (2011)* proposed a neural network approach that integrates the advantages of both machine learning and Information Retrieval techniques. They input semantic orientation indexes into a back-propagation neural network and observed enhanced performance in sentiment classification, along with significant reductions in training time.

- **Logistic regression:** Logistic regression employs maximum-likelihood estimation to determine optimal parameters. The predictor variables can encompass various types, such as continuous, discrete (ordinal and nominal). According to the LR model proposed by *Hamdan, Bellot & Bechet (2015)*, the variable to be predicted (response) is dichotomous in nature, and there is very low correlation between the predictors. In a study carried out by *Saad (2020)*, an extensive Literature review analysis was done to analyze the sentiments on the Twitter data of the US airlines, using six different machine learning models, including naïve Bayes (NB), XgBoost, SVM, random forest, decision tree and logistic regression. Bayes (NB), XgBoost, SVM, random forest, decision tree and logistic regression. To handle the data the researchers used use many pre-processing techniques such as; removing stop words, removing punctuations, stemming and converting to less form. The feature extraction was done using BOW approach using a dataset from Kaggle and CrowdFlower which consists of 14,640 samples belonging to three classes of sentiment namely; Neutral, positive and Negative. The results show that when the dataset is split to 70% for training and the rest for testing the best performance was obtained with support vector machine with accuracy of 83.31% was followed by logistic regression with an accuracy of 81.81%.

### Probabilistic approach

Probabilistic classifiers provide outputs in form of likelihoods on a scale of classes. I want to highlight their easy implementation, comparison of the number of operations with other algorithms, and the ability to work with comparatively small training samples. However, their accuracy of classification might reduce when the nature of data distribution does not match the given assumptions and thus, they may be less efficient than the optimal during such incidences.

Here are some sub-approaches of the probabilistic approach

- **Naïve Bayes:** NB technique can be used for classification as well as for training of samples. It works as Bayesian classification technique which is based on Bayes' theorem. NB operates as a probabilistic classifier that utilizes Bayes' theorem to predict the probability of a set of features corresponding to a particular label. It determines the

conditional probability of an event A occurring based on the individual probabilities of A and B, as well as the conditional probability of event B. Assuming feature independence, NB often uses a Bag-of-Words (BoW) model for feature extraction. Typically, NB is employed when dealing with limited training data. It exhibited a 10% higher accuracy in classifying positive instances compared to negative ones. Despite its simplicity, NB is distinguished by its minimal computational overhead and comparatively high accuracy. The NB classifier employs Bayesian techniques in document classification.

$$p(\text{Class}_i/\text{doc}) = \frac{P(\text{Class}_i)P(\text{doc}/\text{Class}_i)}{P(\text{doc})}. \tag{3}$$

In the realm of text analysis, two predominant models are frequently employed; multinomial naïve Bayes (MNB) and multivariate bernoulli naïve Bayes (MBNB) (*Altheneyan & Menai, 2014*). However, in cases involving continuous data, Gaussian naïve Bayes finds utility as well. MBNB finds application in classification tasks where a dataset is represented by multiple keywords (features).

The NB model introduced by *Tripathy, Agrawal & Rath (2015)* achieved 89.05% accuracy in a K-fold cross-validation setup. This performance surpassed that of other models utilizing the probabilistic NB algorithm (*Calders & Verwer, 2010*). In *Madhuri (2019)*, the author conducted a comparison of four machine learning approaches random forest, SVM, naïve Bayes and C4.5. The dataset, sourced from Twitter and focused on Indian Railways, was categorized into three categories neutral, positive and negative sentiments. Here results indicated that highest accuracy is attained the by SVM which is 91.5%, than random forest attained 90.5%, C4.5 at 89.5%, and naïve Bayes at 89%. *Tiwari et al. (2020)* employed three machine learning approaches, namely SVM, naïve Bayes and maximum entropy alongside feature extraction (n-gram), on the dataset of rotten tomato. Both the training and testing datasets consisted of 1,600 reviews each. The researchers noted a decline in accuracy as the n value increased in n-grams, specifically for values such as four, five, and six. On a different note, *Soumya & Pramod (2020)* using various feature vectors such as BOW and unigram with Sentiwordnet, 3,184 Malayalam tweets were categorized into positive and negative sentiments. They also utilized machine learning approaches like random forest and naïve Bayes (NB). Their findings revealed that random forest achieved superior performance, boasting an accuracy of 95.6%, especially when considering Unigram Sentiwordnet and taking into account negation words. In *Fitri, Andreswari & Hasibuan (2019)*, researchers conducted a sentiment analysis on public comments regarding the Anti-LGBT campaign in Indonesia, using the naïve Bayes algorithm for its high accuracy. The analysis, performed on Twitter data, involved stages of preprocessing, processing, classification, and evaluation. Results indicated that most comments were neutral. The Naïve Bayes algorithm achieved an accuracy of 86.43% in RapidMiner, surpassing decision tree and random forest algorithms, which had accuracies of 82.91%.

- **Bayesian network:** The model operates under the assumption of independence among its nodes due to their random nature, yet it also acknowledges their full dependence stemming from conditional relationships. This comprehensive framework effectively

characterizes the interconnections among collection of variables through a joint probability distribution, facilitating seamless integration of additional variables due to its expansive structure. In a study *Wan & Gao (2015)* utilizing naïve Bayes as a sentiment classifier, researchers employed an ensemble approach incorporating NB, Bayesian network, decision tree, SVM, random forest and C4.5 algorithms. Their findings highlight Bayesian Network's superior performance over the other classifiers in individual assessments.

- **Maximum entropy** The Maximum Entropy Classifier functions as a conditional probability model. Unlike the NB Classifier, it does not depend on prior assumptions such as the independence of keywords within the dataset. Rather than using probabilities to set the model's parameters, the Maximum Entropy Classifier employs search methods to find the parameter set that maximizes its effectiveness. After creating the document-term matrix, the training set is characterized based on its empirical probability distribution, as demonstrated.

$$P * (\text{doc}_i, c) = \frac{1}{N} * (\text{doc}_i, c). \tag{4}$$

Maximum entropy models utilize empirical probability distributions to characterize a dataset, aiming to achieve the highest entropy within the constraints set by prior knowledge. The distinctive allocation representing maximum entropy takes on an exponential form, as illustrated.

$$P(C/\text{doc}_i) = \frac{\exp(\sum_i \Lambda_i f_i(\text{doc}_i, C))}{\sum_\theta \exp(\sum_i \Lambda_i f_i(\text{doc}_i, C))}. \tag{5}$$

*Khairnar & Kinikar (2013)* and *Kaufmann (2012)* employed the maximum entropy classifier to identify parallel sentences across language pairs with limited training data. Unlike other models that either necessitated extensive training datasets or relied on language-specific methods, their approach demonstrated enhanced outcomes across diverse language pairs. This advancement facilitates the creation of parallel corpora for a wide range of languages. In 2015, *Yan & Huang (2015)* utilized the maximum entropy classifier for sentiment analysis of Tibetan sentences, leveraging the disparity in probabilities between positive and negative outcomes. *Boiy & Moens (2009)*, to ascertain sentiment across multilingual texts, merged SVM, MNB, and maximum entropy methods. They applied this approach to various types of content, such as reviews, forum and blogs texts, employing unigram feature vectors.

### Ruled based approach

The concept of rule-based classification encompasses any classification method that utilizes IF-THEN rules for predicting classes (*Machanavajjhala & Gehrke, 2009*). Hence, classifiers employing this approach rely on a predefined series of regulations for sentiment classification. Rule-based classifiers provide rapid classification of new instances and demonstrate performance similar to that of decision trees. An additional advantage lies in their ability to mitigate overfitting. However, interpreting these classifiers can become challenging and labor-intensive, especially when managing a large set of rules. For

instance, *Gao, Xu & Wang (2015)* applied a rule based approach to investigate the triggers of emotions in Chinese microblogs. Their system identifies emotions using an emotion model, activating them when specific rule criteria are met to determine the corresponding trigger.

### Decision tree approach

In this methodology, the training data is structured hierarchically, using conditions on attribute values to classify input data into a finite set of predefined categories. These conditions often revolve around the existence or non-existence of specific words (*Medhat, Hassan & Korashy, 2014*). The resulting tree-like structure, known as a decision tree. The internal nodes of the features represent attribute tests, the branches illustrate the outcomes of these tests, and the leaf nodes indicate either class distributions or child nodes (*Han, Kamber & Pei, 2011*). Decision tree classifiers are valued for their simplicity of understanding and interpretation, as well as their capability to handle noisy data. However, they are susceptible to instability and prone to overfitting (*Berry & Linoff, 2004*). The study *Ngoc et al. (2019)* developed a novel model based using C4.5 algorithm, a decision tree approach within the field of data mining, for document level sentiment classification. Their suggested model secured an accuracy of 60.3% on the testing dataset. Numerous other decision tree algorithms exist with similar functionalities.

### Instance based approach

The KNN algorithm is not widely employed in sentiment analysis, yet it has demonstrated efficacy when trained carefully. Its functioning relies on the principle that the classification of a test instance tends to align with that of its next to neighbors. Determining the optimal K value can be achieved through various hyperparameter tuning techniques like randomized search cross validation or Grid search. The determination of polarity can be achieved either by employing hard voting, which relies on the values of the K nearest neighbors, or by using a soft aggregation method to determine the overall polarity. Table 1 summarizes the machine learning techniques discussed in the reviews above.

## Deep learning approach

In recent times, deep learning algorithms have emerged as the frontrunners in sentiment analysis, surpassing traditional methods. Deep learning represents an ANN consisting of three or more layers, designed to manage extensive datasets along with their intricate characteristics like non-linearity and complex patterns. It undertakes automatic feature transformation and extraction, akin to human learning processes, traversing through multiple hidden layers. These advantages have propelled its popularity in sentiment analysis models since 2015. Many deep learning models prefer word embeddings as input features, which can be acquired from text data using techniques like Word2Vec, embedding layer or GloVe vectors. Word2Vec can be trained through methods such as the Continuous Skip-Gram model or CBOW. Common deep learning algorithms encompass LSTM, RNN, CNNs, RecNN, deep belief networks (DBN) and gated recurrent unit (GRU). Sentiment analysis can be performed through various deep learning methods by many authors as shown in Table 2.

**Table 1  Sentiment analysis by using machine learning approaches.**

| Literature | Language | Classifier | Dataset | Accuracy |
|---|---|---|---|---|
| *Ngoc et al. (2019)* | English | C4.5 Algorithm of a decision tree | (IMDb) (Large 2016) | T1 Accuracy = 60.3% T2 Accuracy = 60.7% |
| *Ye, Zhang & Law (2009)* | English | Naïve Bayes, SVM, N-Gram | Travel Blogs | Accuracy = 80% |
| *Saad (2020)* | English | SVM, Logistic Regression (LR), Random Forest (RF), XgBoost (XGB), Naïve Bayes (NB), Decision Tree (DT) | Tweets of 6 different Airlines in USA | SVM Accuracy = 83.31% |
| *Fitri, Andreswari & Hasibuan (2019)* | English | Naïve Bayes, Decision Tree (DT), Random Forest (RF) | Twitter API | NB Accuracy = 86.41% DT and RF Accuracy = 82.91% |
| *Tripathy, Agrawal & Rath (2015)* | English | Naïve Bayes, SVM | Polarity movie review dataset | NB Accuracy = 89.53 SVM Accuracy = 94.06 |
| *Wan & Gao (2015)* | English | Naïve Bayes, SVM, Bayesian Network, C4.5 Decision Tree and Random Forest | airline service Twitter search API | Ensemble Accuracy = 91.7% |
| *Suresh Kumar et al. (2024)* | English | Decision Tree (DT), Support vector machine (SVM) | Twitter Sentiment analysis Accuracy = 92.78% with 91.33% positive | 97.32% negative sentiment rate |
| *Soumya & Pramod (2020)* | Malayalam | SVM, RF, NB | retrieving tweets using twitter API | Accuracy = 95.6% |
| *Sabir, Ali & Aljabery (2024)* | English | Decision tree (DT), KNN, Naïve Bayes, Logistic Regression and SVM | ChatGPT labeled tweet datasets sourced from the Kaggle community | Accuracy = 96.41% |

For instance, *Jian, Chen & Wang (2010)* leveraged a neural network-based model for sentiment categorization, incorporating feature prior knowledge, sentimental features and weight vectors bases. The researchers used Cornell movie dataset. The findings of this study demonstrated that the I-model accuracy is far surpassed that of both HMM and SVM. In 2019, *Arora & Kansal (2019)* developed the Conv char Emb model, designed to handle distorted data challenges while requiring minimal memory space for embedding. They employed a CNN for embedding, which offers a more parameter-efficient feature representation. In 2020, *Dashtipour et al. (2020)* presented a Persian sentiment analysis framework of deep learning. Their study determined that deep neural networks like CNN and LSTM surpassed traditional machine learning algorithms in performance when applied to hotel and product review datasets.

*Cen, Zhang & Zheng (2020)* introduced, employing three deep learning networks to analyze sentiment in IMDB movie reviews, with a balanced dataset of 50% positive and 50% negative reviews. While RNN and LSTM are common in NLP tasks, CNN, typically

**Table 2  Sentiment analysis by using deep learning approaches.**

| Literature | Language | Classifier | Dataset | Accuracy |
|---|---|---|---|---|
| *Ouyang et al. (2015)* | English | Convolutional Neural Network (CNN) | Movie review dataset from rottentomatoes.com | Accuracy = 45.4% |
| | Thai | CNN, LSTM, Bi-LSTM | Thai Dataset | F1 Score = 0.817 |
| *Tyagi, Kumar & Das (2020)* | English | CNN, Bi-LSTM with Glove | Sentiment140 | Accuracy = 81.20% |
| *Hossen et al. (2021)* | English | LSTM and GRU | Self collected | LSTM = 86% GRU = 84% |
| *Uddin, Bapery & Arif (2019)* | Bangla | Long Short Term Memory (LSTM) with Deep Recurrent model | Bangla tweets | 86.3% |
| *Ray & Chakrabarti (2022)* | English | Convolutional Neural Network (CNN) + Ruled based approach SST-1 (Standford Sentiment Treebank) | SemEval Task 4 | Overall = 87% CNN = 80% Ruled based = 75% |
| *Cen, Zhang & Zheng (2020)* | English | RNN, CNN, LSTM | IMDB Movie Reviews Data | CNN = 88.22% RNN = 68.67% LSTM = 85.32% |
| *Thinh et al. (2019)* | English | ID-CNN with GRU | IMDb | Accuracy = 90.02% |
| *Jang et al. (2020)* | English | Attention based Bi-LSTM + CNN | IMDb | Accuracy = 90.26% |
| *Rhanoui et al. (2019)* | English | CNN, BiLSTM with doc2vec | French articles and international news | Accuracy = 90.66% |
| *Cheng et al. (2020)* | English | Mulit-channel(MC)+ CNN +Bidirectional Gated Recurrent Unit Network + Attention Mechanism MC-AttCNN-AttBiGRU | IMDB dataset + Yelp 2015 dataset | Yelp Accuracy = 92.90% IMDB Accuracy = 91.70% |
| *Raza et al. (2021)* | English | Count vectorization, TF-IDF | COVID tweets Work from home tweets Existing datasets (Kaggle) | Accuracy = 93.73% |
| *Alahmary, Al-Dossari & Emam (2019)* | English | LSTM + Bi-LSTM + SVM | Saudi dialect corpus from Twitter (SDCT) | Bi-LSTM = 94% LSTM = 92% |
| *Mukherjee et al. (2021)* | English | NB, SVM, ANN, RNN | Amazon Reviews Dataset | Accuracy = 95.67% |
| *Abimbola, Marin & Tan (2024)* | English | LSTM + CNN | Canadian Maritime law data | Accuracy = 98.05% |

used in image recognition, also proved effective. Here highest accuracy obtained by CNN that is 88.22%, outperforming RNN at 68.64% and LSTM at 85.32%. CNN architecture denotes a specialized form of feedforward neural network primarily utilized in computer vision (*Ouyang et al., 2015*). However, it has recently demonstrated notable efficacy across

various domains like NLPs and recommender systems. Within a CNN, the layers encompass an input output and a hidden layer comprising pooling layers, normalization layers and fully connected layers with multiple convolutional layers. Convolutional layers process the inputs, such as word embeddings in text sentiment analysis, to identify features. Pooling layers then reduce the feature resolution, making feature detection resilient to noise and minor variations. The normalization layer standardizes the output from the previous layer, aiding in faster convergence during training. Finally, fully connected layers play a crucial role in performing the classification task.

*Ray & Chakrabarti (2022)* introduced a rule-based technique for aspect extraction was combined with a 7-layer deep learning CNN model to label each aspect. This hybrid approach thus yielded an accuracy of 87%, which was superior to the individual models, the rule-based method attained 75% and the CNN model 80% accuracy. This model takes a set of inputs while using RNN employing a memory cell. Due to the remarkable ability to memorize sequential data and recognize sequences with the help of memory cells, RNNs are often used in many NLP projects including sentiment analysis (*Sharfuddin, Tihami & Islam, 2018*). The ongoing research as seen in *Donkers, Loepp & Ziegler (2017)* shows a better performance of RNNs if enough data and computation power is available. All the outputs in RNNs depend on all the previous calculations. For example, in the task of predicting the next word, the model uses states of the previous words and their interaction (*Chen, Zhuo & Ren, 2019*). A main problem experienced when deploying conventional RNNs is the problem of the vanishing gradients.

*Hochreiter & Schmidhuber (1997)* presented the solution in the form of the special RNN training variant, which has received the LSTM name and became popular in different fields. There is increased usage of this architecture by many researchers in sentiment classification exercises. Recurrent neural networks –LSTM, combat the gradient vanishing problem by regulating the input data flow through several gates. Being famous for sequentially structured problems such as forecasting time series and understanding language LSTM applies forget, input, and output gates to deal with the long dependencies providing excellent sequence learning and prediction.

*Tyagi, Kumar & Das (2020)* proposed a system where CNN is used in combination with BiLSTM for sentiment analysis on Sentiment140 dataset. This dataset consist of 1.6 million tweets which were labeled as positive or negative and had gone through several preprocessing steps such as conversion to lower case, stemming, removal of duplicate words such as is, the, and, and, *etc*, numeral digits, URLs and @Tweets, and special characters. The hybrid model had several layers; embedding was done using the GloVe pre-trained model followed by a one-dimensional convolution layer, BiLSTM layer, full connection layer, layer of dropout, and a classification layer. As it will be found in the results, the proposed model obtained an accuracy of 81.20% for Sentiment140.

In *Hossen et al. (2021)*, the authors proposed the application of recurrent neural networks (RNNs) for the sentiment analysis of customers' reviews that are collected from the hotel booking platforms. The researchers applied various preprocessing methods, such as lemmatization, stemming, and the elimination of punctuation and stop words, to prepare the data prior to feeding it into two deep learning architectures: These are LSTM and Grwest

recurrent unit (GRU). LSTM architecture had 30 hidden layers while GRU architecture contained 25 hidden layers. There is a comparison with the LSTM and GRU models as the authors state that on the collected set of data these models made 86% and 84% identification, respectively.

One work that refers to such a hybrid architecture was reported by *Thinh et al. (2019)*, who used 1D-CNN in combination with RNN for sentiment analysis. The IMDb was used to assess this model and consists of 50 000 movie reviews which are classified as positive and negative. The 1D-CNN segment used convolutional layers of 128 and 256 filters, also the RNN segment used LSTM, bidirectional long short-term memory (BiLSTM), and GRU with 128 nodes. This analysis showed that the set up involving 1D-CNN and GRU model outperformed the other networks with an accuracy of 90.02 percent in the IMDb dataset.

Hybrid models were designed by the authors in *Rhanoui et al. (2019)* CNN and BiLSTM networks for the purpose of sentiment analysis. They employed data set of 2003 articles and included positive sentiment, negative sentiment as well as neutral sentiments into account into their research. The text data was preprocessed using word embeddings obtained from converter using the pretrained doc2vec model. The architecture of the hybrid model included several layers: a convolutional layer, a max-pooling layer, BiLSTM layer, dropout layer and classification layer. They did so using this model which was trained to estimate diagnostic accuracy by 90.66% on the data set.

Based on ATT, KEAHT (*Tiwari & Nagpal, 2022*) is designed for sentiment analysis. This innovative model used the BERT pre-trained framework that made this model be easily train on small data. Latent Dirichlet Allocation (LDA) together with a lexicalized domain ontology was integrated in order to resolve issues associated with polarity scoring as well as utility-based optimization in sentiment analysis. In addition, for the sake of enhancing its performance, the model connected external knowledge sources such as sentiment network graphs, the distribution of the text's length, the number of words, and high polarity tweets. The proposed model was tested using two benchmark datasets, COVID-19 vaccine and the Indian farmer protests, and achieved testing accuacy of 91% and 81.49% respectively.

LSTMs forecast, categorize, estimate, and even create sequences, and TensorFlow and PyTorch enable implementing them. CNN was found to be better than LSTM and BiLSTM model on sentiment classification of Thai children's tale in study done by *Pasupa & Ayutthaya (2019)*. The superior performance of the CNN's was due to its nature to handle word groups at once while offering a better prediction on general sentiment while being immune to the memorization problem prevalent to other model like the LTE, BiLSTM which have issues when handling long strings. Having this advantage allowed the CNN to further use the word embedding features with F1 score of 0.817. Another undertook a research study with the same foundation (*Uddin, Bapery & Arif, 2019*) of sentiment analysis of 5,000 Bangla language The experiment uses 5,000 Bangla language Tweets are used, utilizing LSTM (*Hochreiter & Schmidhuber, 1997*). They first preprocessed the dataset to exclude punctions or spaces, then they divided the set into training set (80%), validation set (10%) and test set(10%). Indeed, hyperparameters were tuned and the best accuracy of the model was reported to be 86.3% with the five-layer LSTM architecture where each layer has 128 units and where the batch size is 25 together with the learning rate of

### The hybrid CNN and BiLSTMs were enhanced with an

attention mechanism in a study done in *Jang et al. (2020)*. Their experiments used the IMDb dataset of 50k positive and negative reviews. They extracted the text data pre-processing it with Word2Vec pretrained embeddings. The hybrid model was optimized with an Adam optimizer where we included L2 regularization and dropout techniques. The experiment results in this research indicated that the introduced model in this study obtained the accuracy rate of 90.26% in the IMDb dataset, which proved that the attention mechanism is desirable in sentiment analysis.

Bi-LSTM is preferred over normal LSTM when the overall preceding as well as following sequence data is instrumental in providing optimum forecasting or categorization results especially in natural language processing. Bi-LSTM is capable of receiving information from both directions and hence better in recognizing highly complex patterns as compared to other LSTMs which makes it suitable for instance in applications such as named entity recognition, machine translation and sentiment analysis. In study *Raza et al. (2021)*, authors employed a MLP for the task of sentiment prediction of the Tweets related to COVID-19. The data set comprised of as many as 101,435 'Tweets' These 'Tweets' were preprocessed to delete the tags being HTML and other such nondiscriminating alphabet characters and then proceeded to stem the letters and tokenize them. Then for text two method was used that we all use and those are TF-IDF vectorizer and count vectorizer. These features were separately categorized employing an MLM model that applied five hidden layers of activation and employs ReLU. The analysis revealed that the highest accuracy belongs to MLP Model that used the count vectorizer, 93.73%. *Alahmary, Al-Dossari & Emam (2019)* in their articles presented in 2019 also explored the use of LSTM and bidirectional LSTM (BiLSTM) in sentiment analysis of Saudi dialect text. The researchers made use of a corpus of 60,000 Tweets that were positive or negative. The data set was preprocessed, which involve erasing any special characters, punctuations, numbers, and any letters other than Arabic. Text preprocess steps involved are lemmatization and word vectors were obtained from word2vec model. The total obtained dataset was divided into train and test sets in a random manner where 70% of the test cases were assigned for training and only 30% for testing the outcome of our model. The LSTM model and BiLSTM model were trained and tested respectively. The tests revealed that authors receive higher accuracy rate of 94% from BiLSTM than LSTM. The authors *Mukherjee et al. (2021)* presented a broad set of features to identify an explicit negation and offered a specific algorithm for this idea; to verify it, the authors used several machine learning algorithms for Amazon cell phone reviews. this work indicated that integration of Generalized RNN with the proposed negation processing method provided the Generalized RNN a test accuracy of 95. 67% As such it improved on the Generalized RNN with out negation detection by. Using the same approach to another Amazon review dataset also greatly enhanced the overall accuracy.

GRU, a simpler variant of LSTM or RNN, reduces computational complexity and training time with fewer parameters and a combined gate mechanism. This makes it suitable for resource-constrained scenarios. However, it may not capture long-term dependencies as effectively as LSTM or BiLSTM. *Cheng et al. (2020)* proposed MC-AttCNN-AttBiGRU model combines CNN and bidirectional GRU with an attention mechanism, enhancing

text feature extraction for sentiment analysis. Experimental results on IMDB and Yelp 2015 datasets demonstrate superior performance over baseline models. Despite its capability to extract local features and context semantic information, it's noted that additional improvements may be needed for sentiment classification tasks. The model achieves the best classification results compared to other baseline models on the public datasets. This study *Abimbola, Marin & Tan (2024)* devised an innovative approach for sentiment analysis in Canadian maritime case law using CNN and LSTM models, aiming to reveal biases in legal outcomes. The deep learning approach achieved a high accuracy of 98.05%, significantly outperforming traditional methods like SVM, Naïve Bayes and Logistic Regression, which had accuracies of 52.57%, 57.44%, and 61.86% respectively. The findings underscore the promise of deep learning for legal sentiment analysis and its prospective uses in legal analytics and policy formulation.

Above Table 2 which summarizes the deep learning techniques.

## Transformer based approach

A method in natural language processing (NLP) known as the transformer-based approach utilizes neural network structures that utilize self-attention mechanisms for processing input sequences effectively. Models like BERT (*Devlin et al., 2018*), GPT (*Ethayarajh, 2019*), and their adaptations are trained on extensive text datasets to grasp contextual meanings of words or tokens. They demonstrate proficiency in grasping distant relationships and contextual nuances, rendering them applicable across various NLP tasks, such as comprehension, generation, translation, and sentiment analysis. There are different approaches to sentiment analysis utilizing transformer based approaches by authors.

BERT (*Devlin et al., 2018*) is a state-of the-art model, pretrained in an all-words manner designed by Google AI language that models the text through context. By means of pre-training which includes masked language model and the next sentence prediction, BERT is ready for any other NLP tasks, and it always produces highly accurate results. By virtue of its flexibility and performance, it rendered as a crucial instrument for development in NLP research and application.

*Jiang et al. (2019)* proposed Transformer-based memory network (TF-MN) improves previous methods by framing the task as a question-answering process, optimizing context, questions, and memory modules. It combines local attention and global self-attention mechanisms to better capture the emotional content of web comments. To mitigate the impact of irrelevant words on emotion extraction, improved memory networks are employed. Experiments on two datasets demonstrate that TF-MN outperforms the current high-tech models.

*Hasan et al. (2023)* developed a Bangla annotated dataset for sentiment analysis regarding the Ukraine-Russia war, sourced from YouTube comments on Bangladeshi news channels. It includes 10,861 comments labeled as Neutral, Pro-Ukraine, or Pro-Russia. Various transformer-based models were fine-tuned and evaluated, with BanglaBERT achieving the best performance, reaching 86% accuracy and an F1 score of 0.82. Hyperparameter optimization and multiple evaluation metrics confirmed BanglaBERT's superiority over other models.

*Baniata & Kang (2024)* put forward an approach for Arabic sentiment analysis that employs multitask learning technique along with a switch transformer that utilizes a shared encoder to bolster the results of machine learning when the data available is in significantly limited amounts. Because of using MoE approach, the model can accurately deal with the complicated input–output mapping, and long sequences. The proposed model shows strong performance, achieving accuracy rates of 83.91% on the LABR dataset, 84.02% on the HARD dataset and 67.89% on the BRAD dataset. This approach addresses challenges in one-task learning and enhances sentiment analysis for the low-resource Arabic dialect.

*Dhola & Saradva (2021)* convene a comparative analysis to evaluate the performance of four machine learning algorithms in sentiment analysis; LSTM BERT SVM and multinomial naïve Bayes. They used the Sentiment140 dataset including 1.6 million categorized Tweets classified as negative or positive. Having applied the pre-processing methods of stemming, lemmatization, tokenization, and using the stop-word list and without applying them, their research demonstrated that BERT outperformed the other models with an accuracy of 85.4%.

In *Al Wazrah & Alhumoud (2021)*, the two neural models, SGRU and SBi-GRU with word embedding and a novel ASR method to eliminate stop words to meet its goal of mining Arabic opinions. The performance of these models is evaluated relative to LSTM, SVM, AraBERT, and the ensemble variants of the models. It can be seen that the six-layer SGRU and the five-layer SBi-GRU achieved the peak accuracy and the proposed ensemble model outperformed all models with over 90% overall accuracy. This approach represents the first time that these models/ensembles have been used to classify Arabic sentiment. *Tan et al. (2022)* introduced a combined ensemble deep learning model for sentiment analysis, combining RoBERTa with BiLSTM, LSTM, and GRU to record long-range dependencies in text. The model uses averaging ensemble and majority voting for improved performance, and GloVe pre-trained embeddings for data augmentation to address imbalanced datasets. Experimental results show that this approach surpasses state-of-the-art methods, achieving accuracies of 94.9% on IMDb, 91.77% on the Twitter US Airline Sentiment dataset, and 89.81% on Sentiment140. To gauge public sentiment during the Corona pandemic, this study *Singh, Jakhar & Pandey (2021)* analyzes tweets using the BERT model. It examines two datasets: one with global tweets and another with tweets from India. Validation of emotion classification accuracy was done using a GitHub repository. The experimental results indicate a validation accuracy of approximately 94%.

*Kumar et al. (2024)* presented the Double Path Transformer Network (DPTN) for comprehensive review categorization by modeling both global and local information. The DPTN uses a parallel design combining self-attention and convolutional networks, enhanced by gaining-sharing knowledge optimization (GSK) for hyperparameter tuning. The research shows that optimization algorithms and deep learning can effectively manage class imbalances. Experimental results demonstrate the model's high accuracy, reaching 95%. GPT model (*Ethayarajh, 2019*), a creation of OpenAI, introduces a fresh perspective on comprehending textual emotions. While its main emphasis lies in generating text, GPT indirectly aids sentiment analysis by producing responses relevant to context. Noteworthy attributes of GPT include its adaptability, scalability, ability to grasp context, capacity

for generalization, and innovative approach, rendering it a formidable force in natural language processing tasks.

T5 (*Raffel et al., 2020*) is Google's multilingual and flexible text to text transfer transformer that can be used for multiple NLP operations including text condensation, categorization, and translation. It produces state-of-art performance on most NLP tasks and is also quite versatile, which makes it convenient to adapt for any other NLP task. T5 is mainly an Enhanced Encoder-Decoder Transformer model but comes with different architectures such as pre-normalization. They fine-tuned five different sizes of the model running from the small model up to the model with 3 billion, 11 billion of the parameters. They are highly adaptive to work with just more extended unlabeled data at the time of training.

The recently released XLNet (*Yang, 2019*) language model, proposed and implemented by researchers from Carnegie Mellon University and Google is a groundbreaking innovation in NLP. It uses the most current research findings and is built on a vast NLP corpus; it gives industry-leading performance in objectives such as text categorization, text summarization, and language translation. First, similar to the Transformer-XL, XLNet has also been pre-trained with the autoregressive (AR) approach. Unfortunately, many of the AR models such as GPT and GPT-2 work well for generative NLP tasks but fail to capture bidirectional context. Compared to BERT, XLNet is an AR model that reconstructs data from messed up inputs using mask to recover the original tokens during pre-training and then predict the original sentence from the input (*Pipalia, Bhadja & Shukla, 2020*).

*Fatouros et al. (2023)* focused on analyzing the use of large language models, specifically ChatGPT 3.5, for identifying and interpreting the sentiments associated with the foreign exchange market. The study assesses multiple ChatGPT prompts using a zero-shot prompting method applied to a specially selected set of forex news headlines. This research shows the superiority of ChatGPT over FinBERT in the case of sentiment classification by achieving a 36% higher correlation coefficient with the market returns. This shows the possibility of chat GPT improving the sentiment analysis in the financial domains and further research in the same field.

*Kheiri & Karimi (2023)* aims at analysing the application of Generative Pretrained Transformer (GPT) approaches for sentiment analysis with the SemEval 2017 Task 4. It employs three main strategies: scaffold instruction execution with GPT−3.5 Turbo, GPT models retraining, and an innovative embedding classification scheme. The study also presents a comparative analysis of the performance of the GPT-based model to other models or benchmarks. Findings reveal that GPT methods achieve much larger improvements in predictive accuracy by improving on the best methods by 22% in the F1-score. It also discusses special issues related to SA, including proper context recognition and sarcasm identification.

*Miah et al. (2024)* presents an ensemble model consisting of transformers with an LLM for the polarity classification of foreign languages translated to English. Four languages were translated into English *via* LibreTranslate and Google Translate and then evaluated using pre-trained sentiment models. The accuracy rate of ensemble model analysis on the

translated sentences was higher than 86%, thus indicating the need to perform sentiment analysis of foreign languages through translated English sentences.

The Text Enhanced Transformer Fusion Network (TETFN) is a new approach (*Wang et al., 2023*) that makes use of a text-based cross-modal projection to produce a joint multimodal representation. This method incorporates textual data while retaining isomorphism and identification of signal differences between the modalities. The Vision-Transformer is used to obtain a representation of videos that incorporates both global and local features of human faces. Experiments presented on standard datasets prove the model's effectiveness.

As such, *Lossio-Ventura et al. (2024)* supplies a hands-on guide to analysing sentiment with natural language processing (NLP) tools that might reveal bias in published medical journals about.

*Lossio-Ventura et al. (2024)* supplies a hands-on guide to analysing sentiment with NLP tools that might reveal bias in published medical journals about the chronicity of tick-borne diseases. The study is therefore been undertaken based on 5643 abstracts of research articles in science journals from the period of 2000–2021 on chronic Lyme disease. It explains the demonstration of Python code, pre-trained language models, and basic approaches, including SHAP, and Moreover, ChatGPT is also employed as a quick sentiment analyzer. This works benefits from strong analysis by employing these models and a rich set of data to provide fresh ideas on the discussion about chronic Lyme disease among the medical field. It is an important reference source for the researchers and practitioners who want to use NLP in different healthcare areas including sentiment analysis.

Text is one of the most forms of unstructured data out there and a problem that researchers encounter entails determination of polarity in vast customer feedbacks. The Double Path Transformer Network (DPTN) addresses this by modeling for the global and local information for review categorization. *Kumar et al. (2024)* aligns to a parallel structure that integrates self-attention as the bottom-up and convolution network as the top-down mechanism, incorporating the gaining-sharing knowledge optimization (GSK) method to determine the optimal hyperparameters. The model also shows that optimization algorithms and deep learning can adequately handle class imbalances, experiment analysis provides 95% of accuracy.

Sentiment analysis in NLP poses some difficulties, especially when considering low resource languages like Lithuanian. This opinion is coming from the fact that traditional machine learning approaches and classification methodologies are not very effective in most cases. This work *Vileikyte, Lukoševičius & Stankevičius (2024)* entails the use of transformer models on five-star rating-based online reviews in Lithuanian language; the models to be used here include fine-tuned pre-trained multilingual LLMs such as BERT and T5. The fine-tuned models show a high performance level, as testing recognition accuracy is 80.7% and 89.61% for the respective models, which can be considered higher than the commercial analogue LLM GPT-4.

It is also important to note that the study by *Shaik Vadla, Suresh & Viswanathan (2024)* is an attempt at creating a prediction pipeline used for aspect detection as well as sentiment analysis on the review data. For this purpose, it employs words embedding models like

Bidirectional Encoder Representations from Transformers (BERT) and the Text-to-Text Transfer Transformer (T5) to perform the reviews classification to positive, negative and neutral categories. Specializing in green products, the research detects themes to an extent of 92% and 91%. The selected model we establish as the best performing one depends on the accuracy rate of its performance in different models including precision rate, recall rate, F1 score rate, and time consumption. These results demonstrate how BERT can be used in understanding and analyzing customers' reviews, which prove helpful to designers and researchers of products.

Table 3 which summarizes the transformer based learning approaches discussed in the reviews above.

## Hybrid approach

It aims at integrating one or more techniques in an attempt to improve the performance of sentiment analysis.

In *Learning (2020)*, the authors introduced a multi-stage text classification framework that integrates a lexicon-based approach with stacked machine learning models. We also introduced the Sentiment Score in the feature set as a new feature, which was previously decided by a dictionary-based classifier. The stacked ensemble classifier was built using three machine learning algorithms: SVM, KNN, and C5.0, and the two meta-learners: RF and GLM. The findings of the study showed that the RF meta-classifier was superior to the GLM meta-classifier because it presented a better performance with a proposed accuracy of 90.66% for the five-fold cross-validation data and 91.25% for the ten-fold cross-validation data. Furthermore, when compared with the other single classifiers namely SVM, NB, DT, RF, and ME the proposed approach yielded a higher accuracy.

In *Mendon et al. (2021)*, the authors developed a hybrid model that integrated machine learning and lexicon-based approaches for sentiment analysis of Twitter data on natural disasters. TF-IDF and K-means were used for the sentiment analysis while for the topics extraction, pipeline of Doc2Vec and K-means along with Topic modeling technique known as Latent Dirichlet Allocation (LDA) was used. The dataset included 243,746 tweets related to natural disasters in India. Similarity and polarity indices, as well as the identification of the discussed topics in the samples taken from the Twitter population, was used to classify the sentiment of the sample set.

Other works have integrated lexicon or rule-based approaches with deep learning techniques. In *Wu et al. (2018)*, the authors proposed the unsupervised method for opinion target extraction and aspect term extraction. This approach combined chunk-level linguistic rules with a deep learning model to make more logical predictions and generate higher-level aspect representations using a deep GRU network.

Several studies have integrated the machine learning models into deep learning models as indicated in *Kumar et al. (2020b)* and *Srinidhi, Siddesh & Srinivasa (2020)*. In *Kumar et al. (2020b)*, a hybrid model was presented which combines deep learning (CNN) and machine-learning (SVM) framework to address the multi-modal content including both text and image, and uses decision level multi-modal fusion for sentiment analysis at a more refined level. The model consists of four phases: dividing images and text through

**Table 3  Sentiment analysis by using transformer based approaches.**

| Literature | Language | Classifier | Dataset | Accuracy |
|---|---|---|---|---|
| *Jiang et al. (2019)* | English | Transformer based memory network (TF-MN) | Weibo Semeval-2014 Task 4 | 81.87 |
| *Baniata & Kang (2024)* | Arabic | multi-task learning (MTL) + mixture of experts (MoE) technique | HARD BRAD LABR | HARD = 84.02% BRAD = 67.89% LABR = 83.91% |
| *Dhola & Saradva (2021)* | English | SVM + Multinomial NB + LSTM + BERT | Twitter dataset | BERT = 85.4% LSTM = 80% SVM = 76.3% MNB = 76.9% |
| *Miah et al. (2024)* | English | Transformer and LLM like GPT-3 | SemEval-2017 Task 4 Twitter Sentiment Amazon Reviews DEFT 2017 SENTIPOLC 2016 | 86% |
| *Hasan et al. (2023)* | Bangla | BanglaBERT, XLM-RoBERTa-base, XLM-RoBERTa-large, DistilmBERT and mBERT | YouTube data API | 86% |
| *Al Wazrah & Alhumoud (2021)* | Arbic | SGRU + SBi-GRU + AraBERT | Arabic sentiment analysis (ASA) dataset | Accuracy = 90.21% |
| *Singh, Jakhar & Pandey (2021)* | English | BERT | Tweets from entire world Tweets from india | 94% |
| *Tan et al. (2022)* | Arbic | RoBERTa-LSTM + RoBERTa-BiLSTM + RoBERTa-GRU | IMDb Twitter US Airline Sentiment Sentiment 140 | 94.9% 91.77% 89.81% |
| *Kumar et al. (2024)* | English | CNN + Double Path Transformer Network (DPTN) + Gaining-sharing knowledge optimization (GSK) | Amazon data | 95% |
| *Fatouros et al. (2023)* | English | LLM + ChatGPT3.5 + FinBert | Forex pairs like AUDUSD, EURCHF, EURUSD, GBPUSD, and USDJPY | GPT4 = 76.5% |
| *Miah et al. (2024)* | English | Ensemble model, Twitter-RoBERTa-BaseSentiment-Latest, BERTweet-Base-Sentiment-Analysis, GPT-3 | SemEval-2017 | GPT4 = 85.71%, Ensemble = 86.71% |
| *Kumar et al. (2024)* | English | GSK based DPT network | Amazon dataset | Accuracy = 95% |
| *Vileikyte, Lukoševičius & Stankevičius (2024)* | Lithuanian | BERT and T5 | collect from pigu.lt, atsiliepimai.lt, and google.com/maps | BERT = 80.7%, T5 = 89.61% |
| *Shaik Vadla, Suresh & Viswanathan (2024)* | Lithuanian | BERT and T5 | Amazon dataset | BERT = 92%, T5 = 91% |

discretization, text analysis through CNN, images through SVM, and a boolean decision with OR operation in classification of sentiments.

For example, *Srinidhi, Siddesh & Srinivasa (2020)* introduced a new approach to sentiment categorization, which was based on the hybrid model of MaLSTM and SVM. In this approach, hidden representation was learned through LSTM while classification of sentiment was done through the SVM. The model has been tested on the IMDB dataset.

From these studies, it can be concluded that the hybrid solution is more effective than other relevant methods, such as traditional and deep learning methods and methods based on lexicons or rules. Given that each of the aforementioned approaches has its advantages, hybrid models can capitalize on the strengths of all the Methodologies, thus yielding optimal sentiment classification and a better process of addressing emergent data types. These features enable more accurate feature extraction from the input data, improved learning, and sentiments analysis leading to better results in various datasets and uses.

## APPLICATION OF SENTIMENT ANALYSIS

### Business intelligence

Business intelligence sentiment analysis comes with the following advantages within its domain. Sentiment analysis data can be used by organizations to enhance their products, gain better views of customers' opinions, and create new types of approaches to marketing. One of the most common applications of sentiment analysis in business intelligence is to gauge the customer's attitude towards products or the services they provide. However, these analyses except manufacturers, consumers can also use them to assess products and make informed decisions of purchase. For instance, *Han et al. (2019)* its findings to improve organizational products, assess the returned comments from customers, or create unique marketing campaigns.

- **Product analysis:** Sentiment analysis used by the organization to determine the overall customer sentiments about the organization's products or services based on the comments made on social media sites such as reviews, ratings, and comments. The approach makes it easy for businesses to identify trending products, understand customer issues and improve on customer experience (*Paré, 2003*). Sentiment analysis can be used to measure customer's sentiment concerning a newly introduced product by evaluating the comments made by customers. It is possible to select keywords associated with specific characteristics of a product, for example: food, service, or cleanliness (*Mackey, Miner & Cuomo, 2015*), which makes it possible to build the sentiment analysis framework that is aimed at the identification and analysis only of the specified information.

- **Market analysis and competitor research:** Businesses evaluate opinions of the public concerning these trending topics in order to arrive at decisions concerning the proper product launches and marketing campaigns. Even the opinion of customers about the new features or products can be analyzed using sentiment analysis to improve product and service. Using sentiment analysis, the organizations can track the competitor brands and know about the strength and weaknesses of the competitor brands.

## Capital market prediction

Possible usage of sentiment analysis is a case of predicting capital market trends. This consists of analyzing news associated with the capital market as a way of making price predictions. Examples of data sourcing includes use of data from twitter, news articles and blog posts. From these texts, it is possible to perform the sentiment analysis on the sentence level, and then define how positive or negative toward a particular company. Researcher *Xing, Cambria & Welsch (2018)* has explained that positive news usually has an upward tendency while negative news has a downward tendency. Furthermore, Bitcoin and few other funds relate to Blockchain a magnificent invention that is taking the world by storm.

In the field of stock exchange predictions, sentiment analysis can be effectively applied to assess news articles, financial statements, and, or social media to predict the changes in stock prices, or even more—to predict tendencies in stock exchange. It could be useful for investors to know the current state of feelings of other investors toward a specific company or industry so as to make the right decision either to invest in a particular stock or to sell shares in a specific business.

## Reviews analysis

Sentiment analysis is turning out to be a popular feature especially among entertainment industry. Here it will be possible to examine the reviews of films, television shows, and short films in order to analyze the reaction of the audience (*Kumar, Yadava & Roy, 2019*). This process does not only help the viewers to make decisions, but also serves to bring to the foreground high-quality content. In this domain, the approach suggested in *Lin & He (2009)* that the general sentiment at the level of sentences is useful in dealing with the basic form of evaluations. There are many types of reviews, these include the type of product, service, social, entertainment, health, education, financial, businesses and politics among others.

- **Customer reviews and aspect based:** Over all evaluation of reviews used for non-hotel services such as hotels and restaurants is helpful to customers while making their choices and also assist owners in the refined of their services (*Sann & Lai, 2020*; *Al-Smadi et al., 2018*). Aspect-based sentiment analysis presents the business with a way of making use of the large volumes of data produced. This approach helps organizations locate and prioritize the features of the customer feedback and service. Here ABSA helps to determine the features that are most and least favorable, which can be of great importance for improving the hotel's performance. The opinion identified in the sentiment analysis by *Zhao, Xu & Wang (2019)* points to this industry as being particularly attractive. Thereby the ideas presented can help service providers to target areas that cause customers to provide negative feedback, and thus make leaps and bounds advances in the circle of customer satisfaction.

## Health care

To quantify the satisfaction purposes, the sentiment analysis is used in the operation of the health care industry where patients are able to express their experiences concerning the hospitals, the treatments, and the physicians. This particular analysis is vital in improving

the quality of the level of services offered to the patients. About sentiment analysis mentioning of the topics on social media or transcripts of therapy sessions positive trends in mental health can be detected or indicators of unpleasant emotions. It can also be used on other forms for the evaluation of standards and the new trends in the medical profession as a science. Industry practitioners are currently exploring other uses of sentiment analysis and other NLP methods in the domain (*Ebadi et al., 2021*). As a supplementary data source in public health analysis, the authors of *Clark et al. (2018)* used the Twitter tweets regarding the patient experiences. Using the Twitter's Streaming API. Subsequent to pre-processing, the tweets were labelled using a standard logistic regression classifier as well as a CNN. The research findings identified relationships between positive treatment experiences, the community, as well as awareness. Finally, I opine that the integration of sentiment analysis on the generated content by patients on social media platforms acts as a valuable source of understanding their needs and views.

# CHALLENGES OF SENTIMENT ANALYSIS

## Multilingual sentiment analysis

The main issues arising from multi-lingual sentiment analysis are an indication of the problems posed by multi-lingual sentiment analysis for inherent reasons of working on multiple languages (*Manias et al., 2023*; *Thakkar, Hakimov & Tadić, 2024*). The grammar, lexicon, and even the manner of expressing feelings vary significantly across languages, which hinders developing models that may generalize to different languages. However, there are also certain distinctions peculiar to the manifestation of sentiment, such as cultural differences that may be caused by the fact that certain phrases or words can elicit different sentiments in one language than in another that must also be considered as fundamental. For example, sarcasm, idiomatic expressions and cultural references do not have direct machine translations often and therefore, the models cannot correctly interpret the sentiment. The selection of languages with little labeled data is another significant problem. While languages such as English or Spanish may have plenty of training data available, languages featured by limited speakers often yield low testing accuracy because of a lack of adequate data. Additionally, while using the machine translation to overcome this problem, translation errors or loss of context in translation can hamper sentiment analysis.

## Code-switched sentiment analysis

Code-switched sentiment analysis presents further challenges in their quest because the models have to analyze texts where speakers can flip between two different languages, and this can happen in the middle of a single sentence. This practice is fairly widespread in various multilingual societies; one issue this raises is the language identification issue: which parts of a string belong to which language? Standard sentiment analysis models, while capable of dealing with only one language at a time, fail to handle such transitions. Aside from language identification, it is also important to know how sentiment is expressed when two or more languages intermix. Code-switching can change the connotation of a particular word or phrase in a specific context and emotional valence can alter when

languages are switched (*Ranjan, 2023*). There is also a dearth of very big annotated corpora with code-switched language pairs that are also of high quality. Some of the datasets could be language specific like Hinglish and Spanglish and, much less likely to contain combinations of the specific language pairs that the model is designed to predict (*Almasah, Ebrahim & Abdelaal, 2023*). These challenges make code-switched sentiment analysis to be a tough and technical approach to accomplishing sentiment analysis.

## Cross-domain sentiment analysis

Cross-domain sentiment classification (CDSC) entails developing a model that leverages knowledge from one domain to enhance performance in another. The absence of adequate aspect information can complicate the accurate assessment of sentiment associated with a target phrase, resulting in a considerable number of errors in sentiment classification. For example, the sentence "My friends' car has great look, but engine quality is very poor" here positive regarding the look and negative regarding the engine quality. Historically, the categorization of sentiment based on specific targets has depended on feature engineering to improve classifiers such as SVM (*Wagner et al., 2014*). However, this method is frequently prone to errors and can be quite time-intensive. While sentiment lexicons have been developed to mitigate these issues, their adaptation to new domains continues to pose significant challenges. Recently, advancements in deep learning, including neural attentive models and weakly supervised latent Dirichlet allocation (wsLDA), have been introduced to enhance aspect-level sentiment classification across various domains (*Taboada et al., 2011*; *Yang et al., 2019*). Nevertheless, obstacles in cross-domain sentiment classification remain.

- **Transfer of sentiment features:** Cross-domain sentiment categorization requires the differentiation between pivot and non-pivot elements. Pivot elements, such as "sad", "happy", "bad", and "good", maintain stable sentiment orientations across various domains, facilitating sentiment comparison. Conversely, non-pivot elements are specific to particular domains, like "sweet", and are designed for precise sentiment classification within a specific context. To enhance cross-domain sentiment classification, *Fu & Liu (2021)* introduced a hierarchical attention network known as KPE-net.

- **Multi source:** Sentiment analysis data is frequently sourced from various domains, and conventional domain adaptation methods face challenges in identifying pertinent sources and minimizing negative transfer, which can impair model performance. To tackle this issue, *Fu & Liu (2022)* proposed a contrastive transformer-based domain adaptation (CTDA) approach that employs a mixed selection strategy to identify the top-k sources and utilizes an adaptor to extract information that is invariant across domains.

- **Features identification:** In the statements, "The pizza is very delicious but service is very poor", the term 'pizza' is interpreted positively in one instance and negatively in the other due to service, a challenge that deep learning models frequently encounter. To tackle this issue, *Tang et al. (2021)* introduced the graph domain adversarial transfer network (GDATN), which utilizes labeled data from one domain to infer sentiment in another domain that lacks labels. This approach incorporates a graph attention

(GAT) mechanism and a BiLSTM network, along with a gradient reversal layer (GRL), to identify shared features across different domains, thereby enhancing cross-domain sentiment classification.

### Sarcasm

Sarcasm detection is crucial for NLP applications, such as sentiment analysis and opinion mining, mainly when aggregating students' feedback on course layouts and educational spaces. The article reviewed in *Joshi, Bhattacharyya & Carman (2017)* offered an account of prior work on automatic sarcasm detection and identified key issues that have not been sufficiently addressed. Since sarcasm is an indirect form of speech, then these clues must be deduced by observing the context of prior and proceeding conversations. The research classified sarcasm detection datasets into categories such as short text, long text, transcripts, dialogues, and others, and emphasized three primary methodologies for detection: There are rule-based, statistical, as well as deep learning methodologies. Automatic sarcasm detection involves using of RNN models and LSTM techniques (*Kumar et al., 2020a*), and these can be used singularly or in combination with CNNs. In this concern, the review article shed ample light on these and other methods that can be useful in understanding sarcasm.

### Ambiguity

Historically, indeterminacy is supposed to depend on context and can be interpreted in different ways by different people, and thus poses significant challenges to machine learning algorithms. Such inconsistency can be attributed to structural, syntactic or lexical causes (*Jusoh, 2018*). Whereas structural ambiguity is the phenomenon where a sentence is analyzed through distinct schemes of syntax, syntactic ambiguity is the uncertainty inside particular sub sentential pieces. Lexical ambiguity occurs when a word possesses several meanings or when two words have identical forms, creating difficulties in analyzing feedback.

### Negation

However, negation intensifies the challenges associated with the sentiment analysis solution in terms of its accuracy and usefulness. One of the essential questions is to define to which extent negation operates, whether it is a word, phrase or the whole sentence (*Rahman et al., 2023*). This is made more difficult where there are double negatives which when computing sentiment polarity, one has to be very considerate of when interpreting them. Furthermore, simple use of negation words like not, never and neither adds more complexity on the structure making recognition and handling of them quite exhaustive. Of course there are issues with uncertainty, such as when something is described as "not bad," one cannot easily categories it as good. Included nested and domain specific negations add another level of complexity here as they need context to be identified and handled appropriately. As *Kaur & Sharma (2023)* showed, machine learning models have difficulty with fine distinctions, including negations and other complex structures, indicating that work continues apace. These one need the determined approach based on the promotion of the natural language processing, in context-dependent models and machine learning algorithms improvement.

## Handling emojis based text

Emojis facilitate feelings' expression and clarify the text's meaning to increase the user's comprehension, but their use contributes to communication interpretation problems resulting in inefficiencies. One of the biggest issues for emojis and emoticons for processing is that they usually come in an unformatted form and irregular manner (*Bai et al., 2019*). There are over 3,000 unique emojis, and the authors demonstrated that text pre-processing and dedicated emoji-processing tools are important for obtaining a good sentiment analysis. Emoticons and special characters are essential in opinion mining, especially regarding student feedback, as they convey emotions. Natural Language Processing (NLP) encounters difficulties in accurately processing and tagging these emoticons with appropriate emotional labels. A study *Imran et al. (2020)* examined cross-cultural responses to COVID-19 on Twitter, employing LSTM models and feature extraction methods such as GloVe and word embeddings. The study validated six emotions joy, surprise, sadness, anger, fear, and disgust—through the use of emoticons and their corresponding unicodes.

## Other challenges

- **Data imbalance:** Data imbalance characterized by a significant disparity in the number of samples across different classes (*Thabtah et al., 2020*), poses a prevalent challenge in artificial intelligence, particularly in natural language processing. In the field of education, acquiring substantial labeled datasets is often problematic, as it necessitates expert manual annotation. Furthermore, the application of deep learning algorithms may lead to biased classification outcomes as a result of this unequal distribution of data (*Guo et al., 2018*).
- **Long and complex sentences:** One drawback of many current methods is that the processing of long and complex sentences remains a significant issue (*Bensoltane & Zaki, 2023*). The sort of complexity exhibited in this case is challenging for traditional deep learning models such as the CNN and the GRU. Then LSTM, capsule network, and attention-based models are suitable for such sentences. However, additional development of more flexible and adaptive approaches would make a substantially beneficial impact to elaborate and complex long-sentences to be processed.
- **Idoms and comparative sentences:** The reader may find the ideas explained beyond the understanding since sometimes the meaning is completely different from the literal sense of the words used especially in idioms. For the same reason, comparative sentences also raise the question of ambiguity, as comparative words do not always suggest an opinion. This is so because knowledge graphs are beneficial in studies involving concepts, words, or emotions and their interactions. Besides, sarcasm and irony in most of the times employ the opposite of the opposite of the intended tone and where the use of negation whose intended tone can only be deciphered with a high level of skill (*Tahayna & Ayyasamy, 2023*).
- **Multi-model sentiment analysis (MSA):** Multimedia content from websites serves as a crucial resource for multi-modal sentiment data, with social media providing extensive datasets for examination. Nonetheless, the quality and context of this data can differ significantly and is frequently confined to particular online demographics.

Given its public accessibility, crowdsourcing methods can be employed for classification purposes. Research in MSA indicates that individuals are more likely to share distinctly positive or negative sentiments online, resulting in a scarcity of neutral opinions within MSA studies.

- **Informal writing:** Informal writing presents a considerable obstacle for various NLP tasks, particularly sentiment analysis. People often write informally in their reviews or messages and use abbreviations, icons, and other shortcuts that are difficult for models to decode. While well-known standard acronyms can be addressed, many other regional acronyms are ambiguous and, therefore, changing frequently, which makes it difficult to monitor them and incorporate them into further analyses. Increasing accuracy while preserving computational cost can be achieved by increasing the size of the training sample and the complexity of the model which can result in very high costs. Acronyms can be handled, numerous regional acronyms are continually changing, complicating their tracking and processing.

- **Grammatical mistakes:** Grammatical mistakes frequently occur in informal writing and can be rectified, albeit to a certain degree. While some of the spelling mistakes can be made right, it is still challenging to address unique or recurrent spelling challenges. Erroneous perceptions of smell could further bring improvements in the accuracy of sentiment analysis and other natural language process issues. Languages go through various evolutions as and when they move across regions with factors like phonetics, standard literacy, or historical values defining its features. Although widely recognized acronyms can be handled, numerous regional acronyms are continually changing, complicating their tracking and processing.

- **Computational costs:** Enhancing accuracy typically involves expanding the size of the training dataset and increasing the complexity of the model, which can lead to substantial increases in computational expenses. Training larger models often demands advanced GPUs. In contrast, models such as SVM and Naïve Bayes are comparatively economical, whereas neural networks and attention-based architectures generally incur much higher computational costs.

- **Languages develop and changes:** Languages develop and change as they traverse various geographical areas, shaped by elements such as pronunciation, literacy levels, and cultural significance. For instance, English has a lot of regionalisms, where sometimes a word and its spelling may be completely different depending on the area (*e.g.*, color as opposed to colour). Such lexical variations can lead to situations when an NLP model is both less accurate and computationally costly as it has to cover all the variations of a certain word. Moreover, even though there are thousands of languages, NLP resources are mostly limited to a few languages with most of the focus being on English only.

To sum up, there are emerging challenges facing the sentiment analysis undergoing different forms of data hindering reliable accuracy for similar data. There include multilingual processing. In this processing, the model needs to work on multiple languages. Cross domain adaptation is challenging when the models are trained with the specific type of data since it reduces the model's ability to adapt to a different domain. Sarcasm and

idioms are more sophisticated features, as while sarcasm implies the opposite of the stated emotion, idioms are known as non-literal meaning. Presence of elaborated uncertainty and negation intensify the problem, when words can bear a different meaning depending on a context or when phrases containing a negation change the whole sentiment. Introducing handling of emojis adds another layer, and emojis typically serve or replace certain types of emotional undertones in online, especially casual language. Moreover, quantity and structure of produced text in the form of long and complex sentences, informal language and imbalance of data in training sets also affects the model. Finally, combining text with images or audio during analysis is also not easy, especially when the models used incorporate different inputs differently. Altogether, these make it challenging to conduct sentiment analysis since the challenges posed turn in be complex that requires the use of sophisticated models and approaches.

## EVOLVING TRENDS AND ADVANTAGES IN SENTIMENT ANALYSIS

### Evolving trends

The advancements in the last five years of using sentiment analysis underpin the reinforcement of enhanced machine learning techniques besides the classic machine learning methods; deep learning architectures like RNNs and transformers are intensive for achieving accurate and contextual analysis. The current trends show continued focus on emotion identification which is useful in identification of particular emotions like joy, rage, and sorrow from text information. ABSA has gained interest recently, providing detailed information about certain characteristics of products and services. Similarly, the emerging phenomenon of globalization has focused the need to extend the capability of sentiment analysis across various languages that customers may use to express their feedback towards a given organization. Real-time sentiment analysis tools are being implemented to monitor customer sentiments on the social media sites, and verbal experiences are also tapped into *via* voice and speech integration. It is also applied in market research by the business where they want to have a chronological look at the behavior of consumers towards their brands. Moreover, ethical issues and bias mitigation for filter sentiment models tend to occupy a lot of the focus as well. Last but not the least, the sentiment analysis has now being applied in the less traditional areas including health, laws and policies among many more making it more relevant and dominant.

Ultimately, the change has shifted from traditional approaches applied in the field of machine learning to complex transformers, improve the accuracy and analysis of context. Recent developments include the aspect based sentiment analysis for better feature level experience and real time sentiment analysis over different social media channels so that the feedback received is real time feedback on the customer sentiment. Also, sentiment analysis is very popular in vital sectors like ecommerce platforms, healthcare and policies to measure the perception about market product, health laws and policies. It is therefore important to understate the evolution of sentiment analysis to accommodate the dynamism of language as well as the dynamic user generated content.

### Advantages

- Relative to other related approaches, sentiment analysis gives deeper understanding of customers' satisfaction, choice and complain so as to enhance their experiences.
- It enables one to track live the social media, the reviews and other content which will assist organizations to answer to the public opinions and also manage crisis situations adequately.
- Automated sentiment analysis consumes various text documents in a relatively short time as compared to manual sentiment analysis.
- It can cut down costs of a conventional client data capture mechanism such as sampling through surveys or focus group discussions by leveraging on data mining.
- The sentiment analysis applied on text collected from social media platform assist organizations to make good decisions especially on business strategies, products and services to offered to clients and services to be offered by relevant departments.
- Sentiment analysis in constant provides the competitive advantage as it enables the firm to quickly change to fit the Consumers' expectation.
- It helps in reinforcing brand reputation management by learning changes in consumer attitudes that can be tackled in advance.
- The results of the user feedback affect the subsequent development of the product, which enhances and develops new opportunities for the customer's experience and needs.

## PERFORMANCE EVALUATION PARAMETER

Assessing the performance and efficacy of a methodology or model involves the use of various metrics. This final phase of model development holds particular significance, as not all metrics are applicable to every problem. Sometimes, novel evaluation metrics are introduced to assess newly proposed approaches, as demonstrated in *Jiang et al. (2016)* work. The selection of metrics can significantly influence how a model's performance and effectiveness are gauged and compared. A fundamental confusion matrix typically takes the form of a 2 by 2 matrix, as exemplified in Fig. 4 summarizing the classifier's correct and incorrect predictions. Modern sentiment analysis methods often rely on accuracy, F1 score, and precision as primary evaluation metrics. However, a recent review on sentiment analysis employing deep learning architectures highlights the utilization of recall and accuracy for assessing performance. These metrics are outlined below:

True positive (TP): The number of positive reviews that have been correctly classified.
True negative (TN): The number of negative reviews correctly classified as negative.
False positive (FP): Number of incorrectly classified positive review.
False negative (FN): Number of incorrectly classified negative review

- **Accuracy:** Accuracy, often employed in classification tasks, calculates the percentage of accurate predictions out of the total instances. Error, the inverse of accuracy, can be computed as 1-accuracy. Accuracy is particularly suitable for sentiment analysis in machine learning when there's a relatively even distribution of classes within the dataset.

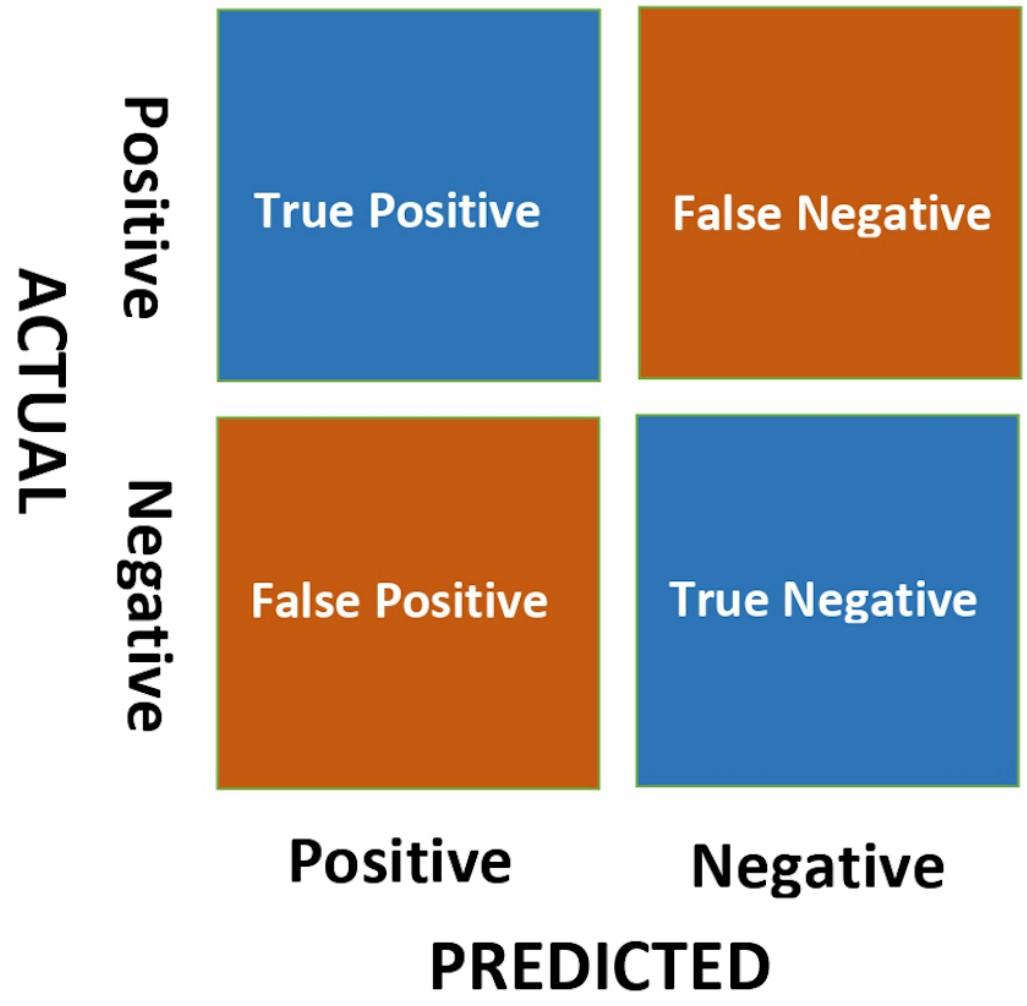

**Figure 4** Confusion matrix showing true positives, false positives, false negatives, and true negatives for binary classification.

$$ACC = \frac{TN + TP}{TN + TP + FP + FN}. \tag{6}$$

- **Precision:** Precision measures the proportion of true positive instances among all instances classified as positive. Essentially, it gauges the accuracy of precise predictions. This measure is applicable to scenarios where confident predictions are essential.

$$PRE = \frac{TP}{TP + FP}. \tag{7}$$

- **Recall:** Recall, also known as sensitivity, signifies the ratio of correctly identified positive instances to all actual positive instances. It assesses the model's ability to capture all relevant instances, focusing on minimizing mis-classifications. Unlike precision, recall

is suitable for tasks where capturing all instances of a particular class is paramount, such as predicting depression. In such cases, predictions need not be overly confident.

$$REC = \frac{TP}{TP + FN}. \tag{8}$$

- **F1-score:** The F1-score, ranging from 0 to 1, provides a balanced evaluation of both precision and recall by computing their harmonic mean. It addresses the challenge of comparing classifiers with high recall and low precision, or vice versa. Hence, the F1-score serves as a solution for this dilemma. It effectively manages the trade-off for models that necessitate confident predictions while also ensuring a dominant capture of a particular class.

$$\text{F1-score} = \frac{2 * \text{Precision} * \text{Recall}}{\text{Precision} + \text{Recall}}. \tag{9}$$

- **Specificity:** Specificity serves as the complement to the recall metric. It is utilized to ensure high precision with confidence in predictions.

$$\text{Specificity} = \frac{TN}{TN + FP}. \tag{10}$$

## FUTURE DIRECTIONS

The future of sentiment analysis is extending in several novel contemporary research trends that focuses on enhancing the efficacy, applicability, and impartiality of sentiment analysis. Two main subcategories concern content-specific sentiment analysis where the models are intended to take into consideration the surrounding text, its individual sentences, and the discourse, in order to capture nuances better. Another area with increasing interest is multichannel sentiment analysis, which combines various types of data including vocal, visual signals and physiological data, including facial expressions, and voice tone to provide more complete picture of the emotions expressed through various channels.

Also, relatively newer deep learning architectures such as transformer-based models, graph neural networks, and reinforcement learning are being used to extended the limits of current components of sentiment analysis, including capturing long-term dependencies and relative connections between various words and phrases. Real-time sentiment analysis is also rapidly developing targeted at the constant tracking of social media, customer opinions, or live data feed for real-time decision-making.

Other interesting areas consist of zero and few shot learning, where GPT or T5 does sentiment analysis without requiring little retraining, and no need for massive annotated data. Cross-lingual and multilingual sentiment analysis is also getting important; models can recognize multiple languages without language-specific data, which makes sentiment analysis more available worldwide. The drive for ethical and XAI with an objective of

developing optimally fair non-biased models that offer easily understandable predictions. Last but not least, personalized sentiment analysis is a relatively new field that takes into account the user's preferences as well as the type of communication and cultural background which generally provides more accurate sentiment predictions taking into account the user's profiles.

Combined, these future directions are the reason why sentiment analysis is becoming more effective, adaptable, and pervasive for various use cases.

## CONCLUSION

In conclusion, sentiment analysis has largely become an essential research topic in the field of natural language processing because of many applications. Unlike many survey articles in this field, this study did not have a purely original approach to the topic in that it did not focus on a particular sentiment classification method. Rather, it reviewed various techniques in sentiment classification and revealed which techniques are optimal for data of different sorts. This approach offered a broader and much more informed outlook on sentiment analysis to what was previously available.

The article examined five key approaches: Four categories which are, lexicon-based, machine learning-based, deep learning-based, transformer-based, and hybrid methods. They also discussed how assessment criteria could be standardized so that the results of one study could reliably be compared to another. The primary focus was on the techniques and issues arising in the execution of sentiment analysis on various data forms, as well as an updated review of the studies dedicated to the treatment of these issues.

Furthermore, the article gave an account of a taxonomy of all the types of sentiment analysis approaches, recent applications, and datasets. Every aspect of the task was discussed, including data preprocessing, sentiment levels, feature extraction, and feature selection, text embeddings, performance indices, and the implementation structure. It also pointed out current shortcomings, explored issues and included recommendations for future research in the article.

Some of the challenges that are associated with text based sentiment analysis include lack of coherence, negation, intensifiers, and semantic meaning. Therefore, current studies seek to optimize previous approaches or propose new solutions for such questions. Moreover, issues like handling multimodal data, joining multiple data sources, and analyzing live and feedback data show us the necessity to investigate new and unconventional computational methods. This is important coming at a time the scientific community is seeking to enhance the accuracy of sentiment analysis of social media and other related fields.

Key gaps in common SA approaches provide insights into the further development of SA in several essential aspects. Another potential issue that has not been well addressed to date is understanding jokes or idiomatic expressions. Language identification and the management of code-switching also continue to pose a challenge, since models have to be compatible with languages and dialects. There is also no information on how to transfer the sentiment analysis to other fields; the models learned with one data type (for instance, customer reviews) are poor when applied to others, including legal or medical. Further,

when working with multimodal data, by which is meant combining text with images, audio or video, it might be possible to obtain deeper sentiment insights, especially for applications in the social media and other interactive usage scenarios.

Future work on sentiment analysis should aim at designing better transformer based models that are more contextual for multiple languages and domains, start with better general models and extend their applicability to low resource languages and to real-time multimodal analysis. Such enhancements will sufficiently solve present deficiencies and further advance sentiment analysis solutions across the entire world.

## ACKNOWLEDGEMENTS

I am deeply grateful to my family for their unwavering support, to my classmates for their valuable insights and collaboration, specially to my mentor (DR. Lal Khan) for their guidance, and to the memory of my late father, whose inspiration continues to motivate me.

### Funding
This work was supported by the National Science and Technology Council (NSTC) of Taiwan under grant number 112-2410-H-182-026-MY2 and Chang Gung Memorial Hospital under project number NERPD4N0232. The funders had no role in study design, data collection and analysis, decision to publish, or preparation of the manuscript.

### Grant Disclosures
The following grant information was disclosed by the authors:
The National Science and Technology Council (NSTC) of Taiwan: 112-2410-H-182-026-MY2.
Chang Gung Memorial Hospital: NERPD4N0232.

### Competing Interests
The authors declare there are no competing interests.

### Author Contributions
- Mahander Kumar analyzed the data, prepared figures and/or tables, authored or reviewed drafts of the article, and approved the final draft.
- Lal Khan analyzed the data, prepared figures and/or tables, authored or reviewed drafts of the article, and approved the final draft.
- Hsien-Tsung Chang analyzed the data, prepared figures and/or tables, authored or reviewed drafts of the article, and approved the final draft.

### Data Availability
This is a literature review.

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
