# Peer review of "Evolving techniques in sentiment analysis: a comprehensive review"

_PeerJ Computer Science, doi:10.7717/peerj-cs.2592_

## Round 0.1 · original submission · Major Revisions

Dear authors,

Thank you for submitting your literature review article. Reviewers have now commented on your study. Your article has not been recommended for publication in its current form. However, we encourage you to address the concerns and criticisms of the reviewers and to resubmit your article once you have updated it accordingly. When submittin the revision, please provide a clearly defined research question for this literature review paper. Furthermore, the Abstract should be attractive and contain motivation. Adding a comprehensive discussion for synthesis of findings, implications, future research, and limitations will be better.

Equations should be used with equation number. Explanation of the equations should be checked. All variables should be written in italics. Definitions and boundaries of all variables should be provided. Necessary references should also be given. "Percision" in the equations must be corrected.

Best wishes,

Reviewer 1 ·

Basic reporting

Is the review of broad and cross-disciplinary interest and within the scope of the journal?
No. The article’s focus on sentiment analysis, but some how the emotion detection interfere in the middle of the content.

Has the field been reviewed recently? If so, is there a good reason for this review (different point of view, accessible to a different audience, etc.)?
Yes, the field has been reviewed recently. However, this review does not present significant differences or a unique perspective compared to other recent reviews in the domain.

Does the Introduction adequately introduce the subject and make it clear who the audience is/what the motivation is?
No. The stated contribution of the survey to provide the trends, advantages and challenges, were missing.

Experimental design

Is the Survey Methodology consistent with a comprehensive, unbiased coverage of the subject? If not, what is missing?
No. The methodology for selecting articles is not sufficiently clear, and the inclusion criteria seem limited to specific terms (e.g., LSTM, BERT, GPT). The authors should consider broader terms like deep learning and transformers, which could encompass a wider range of relevant literature, including CNN, GRU, and other architectures.

Are sources adequately cited? Quoted or paraphrased as appropriate?
No comment.

Is the review organized logically into coherent paragraphs/subsections?
No. The content does not align with the fundamental figure provided in the review. Additionally, some sections are inconsistent.

Validity of the findings

Is there a well developed and supported argument that meets the goals set out in the Introduction?
No. The article lacks a critical analysis of the trends and challenges in sentiment analysis, which are mentioned as the review's main contribution.

Does the Conclusion identify unresolved questions / gaps / future directions?
No. The conclusion fails to identify key gaps or offer future directions for the domain.

Additional comments

1. Please address all the comments and thoroughly revise the article before submitting it to any journal.

2. Recheck the consistency of the terminology, fix any typos, and consider reducing or replacing commonly generated AI terms as in: https://doi.org/10.48550/arXiv.2406.07016

Annotated reviews are not available for download in order to protect the identity of reviewers who chose to remain anonymous.
Cite this review as

Reviewer 2 ·

Basic reporting

Some minor grammatical errors and spelling errors exists. Please proofread the article again for these.

Some literature references are only briefly discussed while others are lengthy. In particular, those under sub-sections 3.4.x can benefit from more discussion.

Figure 3 needs to be more legible.

The evolution of sentiment analysis is not clearly portrayed. This can be done using a timeline.

Please refer to the comments in the attached file.

Experimental design

The review still has gaps to be filled. One perspective which was not discussed is the multi-lingual sentiment analysis and code-switched sentiment analysis, which is quite common nowadays based on the usage of social media by the younger generation who can speak and write more than one language.

Some sections seem disjunct or the sentences in the paragraphs are not directly related to the section header.

Lexicon-based and dictionary-based approaches need to be looked into as they seem overlapping.

Please refer to the comments in the attached file.

Validity of the findings

While there is a wide coverage of sentiment analysis-related students, gaps remain to be filled. Hybrid approaches were not discussed, datasets used were not discussed, challenges and limitations were not discussed. Applications of sentiment analysis were also not discussed.

Future directions and conclusion are briefly discussed. More details should go into what direction can be taken for future works.

Additional comments

Please refer to the above comments as well as the comments in the attached pdf for amendments.

Annotated reviews are not available for download in order to protect the identity of reviewers who chose to remain anonymous.
Cite this review as

·

Basic reporting

The article provides a comprehensive overview of sentiment analysis techniques, covering both traditional methods and emerging approaches like deep learning and transformer models. The literature is well-referenced, including relevant sources from different time periods and methodologies, offering a solid foundation for understanding the evolution of sentiment analysis. Notably, the inclusion of work such as "Analyzing Public Sentiment on Sustainability: A Comprehensive Review and Application of Sentiment Analysis Techniques," co-authored by Tess Anderson, Robert Kelley, and myself, recently published in the Natural Language Processing Journal (2024), demonstrates the growing cross-disciplinary applications of sentiment analysis. The paper is well-structured, with clear sections that guide the reader through the different stages of sentiment analysis, making it accessible to readers with various levels of expertise.

Suggestions for Improvement:
1 Clarify the target audience and provide more context about recent advancements, specifically highlighting why this review is necessary and how it differs from previous literature.
2. Improve the clarity and professionalism of language by avoiding long, complex sentences and informal phrases.

Experimental design

The article content is within the Aims and Scope of the journal, and the review aligns well with the article type. The investigation appears thorough and meets technical standards, but there are a few areas that could benefit from more detail.

While the methods are generally described, they could be expanded to provide more detailed explanations, especially regarding the data extraction and analysis techniques. Including more precise descriptions of how certain techniques were evaluated would help ensure the study is replicable by other researchers.

The survey methodology covers many aspects of sentiment analysis, but it would benefit from further elaboration on how emerging models like BERT, GPT, and transformer-based methods are changing the landscape. This would make the review more comprehensive and up to date. Additionally, more emphasis on potential biases in sentiment analysis datasets and how they are mitigated would improve the rigor of the study.

Sources are adequately cited, and the review is well-organized into coherent sections, though some paragraphs could be condensed to enhance readability.

Suggested Improvement:

Expand the methodology section to ensure replicability and cover emerging models in greater depth. Provide further insights into the mitigation of bias in sentiment analysis models.

Validity of the findings

The article provides a well-developed and supported argument that meets the goals set out in the Introduction. It effectively reviews a wide range of sentiment analysis techniques, aligning with the paper's aim to provide a comprehensive overview. However, the impact and novelty of the findings are not thoroughly assessed. The paper would benefit from a more explicit discussion on how the insights presented contribute to advancing the field or addressing specific gaps in the current literature.

The conclusions are generally well-stated and linked to the original research question, but they could be strengthened by identifying more clearly unresolved questions or areas for future research. While the review touches on emerging trends and techniques, a clearer outline of specific gaps in existing methodologies and potential directions for future exploration would enhance the overall value of the paper.

Suggested Improvement:

Add a section that assesses the impact and novelty of the findings, explaining how the review contributes to the advancement of sentiment analysis.
In the Conclusion, unresolved questions are identified, and specific areas for future research are suggested.

Cite this review as

---

## Round 0.2 · Major Revisions

Dear Authors,

Thank you for submitting your revised Literature Review article. Feedback from the reviewers is now available. The initial submission was deemed to contain numerous issues that remain unresolved by Reviewer 1. It is therefore strongly recommended that the necessary corrections be made and that the paper be resubmitted after the requisite changes have been made.

Best wishes,

Reviewer 1 ·

Basic reporting

Is the review of broad and cross-disciplinary interest and within the scope of the journal?
Yes, the article’s focus on sentiment analysis is within the scope of the journal. However, the scope of the review was too broad, leaving many relevant domains underexplored.

Has the field been reviewed recently? If so, is there a good reason for this review (different point of view, accessible to a different audience, etc.)?
Yes, this field has been reviewed recently, with many similar reviews and surveys published. This particular review does not offer significant new insights or a unique perspective compared to other recent literature in the domain.

Does the Introduction adequately introduce the subject and make it clear who the audience is/what the motivation is?
No. The stated contribution of the survey—to provide an in-depth analysis of text sentiment analysis, including a summary of both advantages and challenges—was not fully addressed.

Experimental design

Is the Survey Methodology consistent with a comprehensive, unbiased coverage of the subject? If not, what is missing?
No. The methodology for selecting articles lacks clarity, and the criteria or outcomes of the article selection process are not provided.

Are sources adequately cited? Quoted or paraphrased as appropriate?
No comment.

Is the review organized logically into coherent paragraphs/subsections?
Yes.

Validity of the findings

Is there a well-developed and supported argument that meets the goals set out in the Introduction?
No. The article lacks a critical analysis of the trends and challenges in sentiment analysis and does not adequately summarize them.

Does the Conclusion identify unresolved questions / gaps / future directions?
No. The conclusion does not thoroughly identify or discuss key gaps in the field or suggest future directions.

Additional comments

Please refer to recent review samples (covering methodology, analysis, challenges, and future work) to enhance the manuscript. Some examples include:

1. https://doi.org/10.1016/j.jksuci.2023.101776
2. https://doi.org/10.1186/s40537-021-00536-5
3. https://doi.org/10.1002/widm.1366

Please also review the flow of writing and proofread carefully before resubmission.

Cite this review as

Reviewer 2 ·

Basic reporting

Authors have addressed all previous comments sufficiently. No further comments from me.

Experimental design

The structure of the paper has been improved with additional sections to fill in the gaps of what were commented to be missing in previous review.

Validity of the findings

Authors have addressed all previous comments sufficiently.

Additional comments

No additional comments.

Cite this review as

·

Basic reporting

Clear and unambiguous, professional English used throughout:
The manuscript is clearly written in professional English with no major language issues.

Literature references, sufficient field background/context provided:
The literature is well-referenced and provides sufficient context. The manuscript covers both classical and recent developments in sentiment analysis.

Professional article structure, figures, tables, and raw data shared:
The article follows a clear structure, with appropriate figures and tables to support the content. Raw data is not applicable.

Scope of the journal and cross-disciplinary interest:
The review is within the scope of the journal and is of broad interest due to its coverage of both traditional and modern sentiment analysis techniques.

Experimental design

Aims and Scope:
The article aligns with the journal's aims and scope, focusing on sentiment analysis techniques and trends.

Methodology and coverage:
The review methodology is consistent with comprehensive, unbiased coverage of the topic. Sources are adequately cited, and the review is logically organized.

Validity of the findings

Well-developed argument and conclusions:
The conclusions are well-supported and align with the research questions. The review identifies gaps and future directions.

Additional comments

No additional comments.

Cite this review as

---

## Round 0.3 · accepted · Accept

Dear Authors,

Thank you for the revised paper. The previous reviewer declined to assess the latest revision, and I have therefore undertaken this task myself. It is my assessment that the paper has been sufficiently improved and is now ready for publication following the second revision.

Best wishes,